# Engineered repeat proteins as scaffolds to assemble multi-enzyme systems for efficient cell-free biosynthesis

Alba Ledesma-Fernandez [1], Susana Velasco-Lozano [1,2,3], Javier Santiago-Arcos [1], Fernando López-Gallego [1,4] ✉ & Aitziber L. Cortajarena [1,4] ✉

Multi-enzymatic cascades with enzymes arranged in close-proximity through a protein scaffold can trigger a substrate channeling effect, allowing for efficient cofactor reuse with industrial potential. However, precise nanometric organization of enzymes challenges the design of scaffolds. In this study, we create a nanometrically organized multi-enzymatic system exploiting engineered Tetrapeptide Repeat Affinity Proteins (TRAPs) as scaffolding for biocatalysis. We genetically fuse TRAP domains and program them to selectively and orthogonally recognize peptide-tags fused to enzymes, which upon binding form spatially organized metabolomes. In addition, the scaffold encodes binding sites to selectively and reversibly sequester reaction intermediates like cofactors via electrostatic interactions, increasing their local concentration and, consequently, the catalytic efficiency. This concept is demonstrated for the biosynthesis of amino acids and amines using up to three enzymes. Scaffolded multi-enzyme systems present up to 5-fold higher specific productivity than the non-scaffolded ones. In-depth analysis suggests that channeling of NADH cofactor between the assembled enzymes enhances the overall cascade throughput and the product yield. Moreover, we immobilize this biomolecular scaffold on solid supports, creating reusable heterogeneous multi-functional biocatalysts for consecutive operational batch cycles. Our results demonstrate the potential of TRAP-scaffolding systems as spatial-organizing tools to increase the efficiency of cell-free biosynthetic pathways.

Cell-free biocatalytic systems are becoming more popular as a substitute for conventional chemical catalysts as enzymes are more sustainable and selective to manufacture valuable chemicals. Applied biocatalysis and more specifically chemical biomanufacturing have benefited from the outstanding advances in molecular and synthetic biology, spurring the creation of novel enzyme cascades[1,2]. Among other approaches, to increase the throughput of enzyme cascades, biotechnologists have designed protein and DNA-based scaffolds to spatially organize multi-enzyme systems within a few nanometers for efficient cell-free biosynthetic pathways. These scaffolded biocatalytic complexes can be tethered to the surface of artificial materials such as porous agarose beads[3,4], cellulose particles[5], or

[1]Center for Cooperative Research in Biomaterials (CIC biomaGUNE), Basque Research and Technology Alliance (BRTA), Paseo de Miramón 194, Donostia-San Sebastián 20014, Spain. [2]Institute of Chemical Synthesis and Homogeneous Catalysis (ISQCH-CSIC), University of Zaragoza, C/ Pedro Cerbuna, 12, 50009 Zaragoza, Spain. [3]Aragonese Foundation for Research and Development (ARAID), Zaragoza, Spain. [4]Ikerbasque, Basque Foundation for Science, 48009 Bilbao, Spain. ✉e-mail: flopez@cicbiomagune.es; alcortajarena@cicbiomagune.es

lipid droplets[6] to heterogenize them, facilitating their separation and recycling.

Among the protein-based scaffolds to organize multi-enzyme systems, cohesin-dockerin interactions derived from natural cellulosomes dominate the landscape of applications[3,7,8]. Nature offers a limited diversity of cohesin/dockerin pairs, which can be assembled as synthetic cellulosomes formed by fusion proteins containing different cohesin domains (scaffoldin) that reversibly bind several enzymes fused to their cognates dockerin domain through calcium-driven protein-protein interactions with extremely high affinity ($K_D$ in nM-pM range)[9]. Besides cellulosome-based scaffolds, other protein domains like affibodies[10], leucine zippers[11], PDZ and SH3 domains[12,13], CipB scaffolds[12], and SpyCatcher/Spy-Tag system[1], have been successfully exploited to organize enzymes in space. Both inside and outside of cells, the spatial proximity of these scaffolded enzymes has proven to be an excellent approach to increase the catalytic efficiency of cascade reactions compared to their non-assembled counterparts where enzymes are diluted in the reaction medium.

Although the overall catalytic performance of multi-enzyme systems assembled on biomolecular scaffolds is enhanced, the fundamentals underlying this improvement are still not understood and thus remain the subject of a vibrant debate. The most accepted theory to explain the enhancement in the performance of scaffolded multi-enzyme systems is the improved mass transport of intermediates between the neighboring enzymes. Owing to the physical proximity of enzyme active sites, the transport of intermediates might be facilitated, resulting on a positive impact on the final rate of product formation[14–16]. This channeling not only benefits the overall kinetics of in vitro cascades but also increases the product titers through reducing the accumulation of labile intermediates in the bulk, avoiding the derailment of intermediates to off-target products and shifting the thermodynamic equilibrium of cascades towards the target product[17]. Hess and co-workers refuted that nanometric proximity of the enzymes avoids the diffusion of the intermediates to the bulk as the diffusion rate of small molecules is normally much higher than the catalytic efficiency of the enzymes. To explain the greater performance of cell-free multi-enzyme systems when scaffolded within biomolecular chassis, it has been proposed the existence of a diffusion-limited effect that poses the system out of equilibrium. Such diffusion-limited effect may take place when the intermediates are reversibly absorbed to a scaffolded architecture (i.e. electrostatic interactions). Herein, higher local concentrations of the intermediates in the environment of scaffolded enzymes may occur, enhancing the efficiency of the system when the second enzyme is the rate-limiting one. To a certain extent, these scenarios can be considered a kind of "intermediate channeling" that enhances the reaction flux towards the target product. The electrostatic guidance emerges as an efficient mechanism for substrate channeling[18], however experimentally it has only been demonstrated using DNA scaffolds[19,20], and by enzyme fusions through positively charged linkers[20,21], and up to date has never been addressed using a protein scaffold.

Aiming at generating efficient multi-enzyme systems, here we propose to exploit not only the precise protein spatial organization of a multi-enzyme system with nanometric resolution, but also encode interactions with the cascade intermediates to increase their local concentration. Toward this aim, we propose the use of engineered proteins to assemble multi-enzyme systems, encoding by design the nanometric proximity of catalytic domains and the binding sites to reversibly sequester cascade intermediates. Although widely used for the fabrication of genetically programmed biomaterials with multiple applications[22], the tetrapeptide repeat proteins (TPR) have not been used as scaffolds to spatially organize multi-enzyme systems. The TPR domain is a 34 amino acid helix-turn-helix repeated motif[23] that enable protein-protein interactions within the cells. TPR domains are easy to engineer, thereby they can be genetically programmed to mediate protein–protein interactions[24]. We have selected three engineered TPR peptide affinity domains (TRAPs), which bind a set of peptides with high specificity and minimal cross-reactivity inspired by HOP TPR units[25]. In addition, the physicochemical properties of TPR domains, including surface charge, can be modified by altering non-conserved, surface-exposed amino acids[26]. As a result, TRAP-based scaffolds offer greater versatility and flexibility compared to cellulosome-based artificial scaffolds, which have limited natural interaction diversity and are not easily engineerable, as TRAPs can be easily redesigned through interface mutations.

Bio-redox reactions coupled to the regeneration of soluble NAD(P)H cofactors are one of the most suitable systems to assess the effects of biomolecular scaffolds on the catalytic efficiency of multi-enzyme systems. These processes work more efficiently when both the main redox reaction and the cofactor recycling step function simultaneously in one-pot as the exogenous cofactor can be supplied at lower concentrations and the thermodynamic limitations due to the underlying reversible nature of the dehydrogenases are minimized. Spatial organization of the participating dehydrogenases is a successful solution to improve the efficiency of in situ NADH recycling[27].

In this work, we exploit engineered TRAP domains to assemble a formate dehydrogenase from *Candida boidinii* (FDH)[28] and an alanine dehydrogenase from *Bacillus stearothermophilus* (AlaDH)[29]. These two enzymes are known to simultaneously perform the asymmetric reduction of α-ketoacids to L-amino acids while recycling NADH using formate as an ancillary electron donor and $CO_2$ as a by-product. We expand this technology using a trivalent TRAP to a tri-enzymatic system including the two abovementioned enzymes plus an ω-amino transaminase from *Pseudomonas fluorescens* (ωTA)[30], for the amination of benzaldehyde with in situ recycling of the electron (NADH) and amine (L-Alanine) donors. NADH channeling between the two scaffolded dehydrogenases was demonstrated, resulting in higher cascade productivity and product titer. Finally, the scaffolded system is immobilized on porous solid particles, creating a multi-functional and spatially organized heterogeneous biocatalyst that can be recycled for a few cycles. This study provides solid evidence that the physical proximity of enzymes through assembly on engineered protein scaffolds, along with selective interactions with reaction intermediates, enables intermediate channeling between active sites and results in enhanced catalytic efficiency.

## Results

### Design of the scaffolding strategy to assemble multi-enzyme systems

Tetratricopeptide Affinity Repeat Proteins (TRAPs) were chosen as scaffolds for enzyme assembly due to their excellent orthogonality in binding different cognate peptides. Initially, we selected two previously designed TRAP1 and TRAP3 modules as they selectively bind the 5-amino acid peptides MEEVV (peptide-1) and MRRVW (peptide-3), respectively (Fig. 1a)[25]. TRAP1/TRAP3 pairs are highly orthogonal as the affinity of TRAP1 toward peptide 1 is 250 times tighter than toward peptide 3, while TRAP3 binds to peptide 3 with an affinity 100 times tighter than to peptide 1[25]. Moreover, the surface-exposed residues of TPR domains that are not involved in the peptide binding are negligibly conserved, which means they can be modified without affecting the structure or recognition capabilities. In particular, the surface of TRAP domains exposed patches of positively charged residues that can be exploited to electrostatically bind phosphorylated cofactors, such as NADH and NAD+. With this knowledge in mind, we fused peptide-1 (MEEVV), and peptide-3 (MRRVW) to the C-terminus of FDH and AlaDH, respectively (Fig. 1a). As FDH is a dimmer, it displays two units of peptide 1, while the hexameric structure of AlaDH displays six units of peptide 3. Once the FDH and AlaDH enzymes are fused to their respective peptides, they are named FDH1 and AlaDH3, respectively. The TRAP1-3 scaffold was generated by the fusion of the TRAP1 and 3

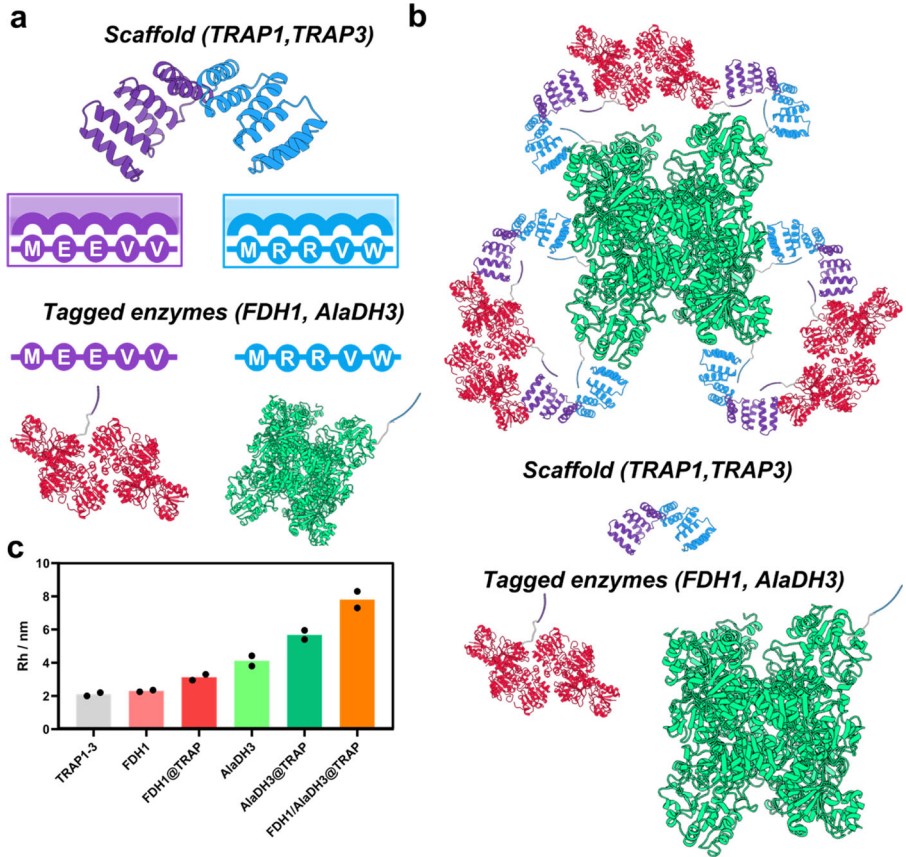

**Fig. 1 | Scaffolding strategy to assemble multi-enzymatic systems. a** Schematic representation of the TRAP-based scaffold composed by TRAP1 (in purple) and TRAP3 (in blue) and their corresponding cognate recognition peptides (MEEVV for TRAP1 and MRRVW for TRAP3) fused to FDH dimer (in red, PDB ID: "5DNA [https://doi.org/10.2210/pdb5DNA/pdb]") and AlaDH hexamer (in green, PDB ID: "1PJB [https://doi.org/10.2210/pdb1PJB/pdb]"). **b** Assembly architecture, in which first AlaDH3 hexamer nucleates 6 TRAP1-3 units and the complex is then loaded with three FDH1 dimers, resulting in a 6:3:1 TRAP1-3:FDH1:AlaDH3 stoichiometry in the final assembly (i.e. a 1:1:1 stoichiometry of monomers). **c** Hydrodynamic radius (Rh) of the different systems measured by Fluidity One W. Rh of free scaffold (TRAP1-3), tagged enzymes (FDH1 and AlaDH3), incomplete assemblies with only one enzyme bound (FDH1@TRAP and AlaDH3@TRAP), and complete assembly with the two enzymes bound (FDH1/AlaDH3@TRAP). The data are presented as the mean of two replicate experiments ($n = 2$). Source data are provided as Source Data file.

binding domains to bind nearby the two tagged enzymes FDH1 and AlaDH3 (Supplementary Table 1). These protein constructs were expressed, purified, and characterized. First, their molecular masses were analyzed by electrophoresis (SDS-PAGE) and mass spectrometry. As expected, based on their sequence, TRAP1-3 scaffold presented a mass of 32,621 kDa, FDH1 a mass of 45,137 kDa, and AlaDH3 a mass of 44,323 kDa (Supplementary Fig. 1a, b). In addition, circular dichroism analysis revealed that the secondary structure of the tagged enzymes was negligibly affected by the peptide fusion and showed that the individual proteins were stable and displayed cooperative thermal denaturation transitions (Supplementary Fig. 1c). The oligomeric state of the enzymes was evaluated by size exclusion chromatography (SEC). The estimated MW by SEC for AlaDH3 and FDH1 were 286 kDa and 95 kDa, respectively, in agreement with the expected hexameric state of AlaDH3 (266 kDa), and dimeric state of FDH1 (90 kDa) (Supplementary Fig. 2 and Supplementary Table 2). Therefore, the tagging of cognate peptides affected neither the secondary nor the quaternary structure of the enzymes. The specific enzyme activity of the FDH and AlaDH was evaluated and compared to the enzymes fused to the recognition peptides (FDH1 and AlaDH3) showing that the addition of the peptides negligibly affects their specific activity (Supplementary Table 3).

Once the scaffolding unit and the tagged enzymes were structurally and functionally characterized, we performed their stepwise assembly in solution. First, TRAP1-3 was mixed with AlaDH3 for 1 h at 4 °C, and then such one-enzyme assembly was incubated with FDH1 under the same conditions to form the target two-enzyme assembly (Fig. 1b). As the assembly was performed sequentially, AlaDH3 hexamers were used as the nucleating unit of the assembly and incubated with TRAP1-3 monomers at a stoichiometry of 1:6 to load all the AlaDH3 subunits with one TRAP unit. Then, three FDH1 dimers were incubated per hexamer of AlaDH3, resulting in an assembly with a theoretical stoichiometry of 6:3:1 (TRAP1-3:FDH1:AlaDH3), which means a 1:1:1 molar ratio, when accounting for the protein monomers. In this compact and well-defined assembly, all the tagged peptides are expected to be bound to their corresponding TRAP module (Fig. 1b). To evaluate the assembly process TRAP1-3 scaffold was labeled with AlexaFluor-647 dye (AF647). The hydrodynamic size of the labeled TRAP1-3 domain, and the changes in the hydrodynamic radius (Rh) upon enzyme assembly was assessed by microfluidic diffusional sizing[31]. The TRAP1-3:FDH1:AlaDH3 complexes showed an increase of their hydrodynamic radius (Rh) compared to TRAP1-3. The Rh of the assembled system was larger than the two non-scaffolded enzymes and the tagged enzymes individually assembled into TRAP1-3 (Fig. 1c). These results demonstrate the efficient assembly of the two tagged enzymes within the engineered scaffold.

Once the assembly conditions were optimized, we analyzed the assemblies by size exclusion chromatography (SEC) to determine the components of the assembly and its approximately molecular weight. The FDH1/AlaDH3@TRAP assembly showed a unique chromatographic peak indicating the absence of free elements and the efficient engagement of TRAP scaffold in the interactions with the different

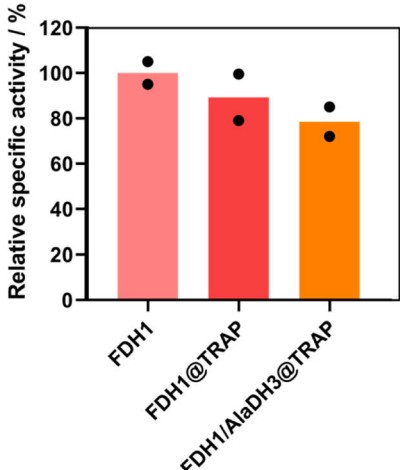
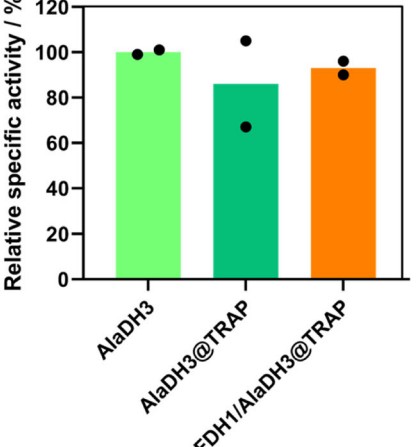

**Fig. 2 | Activity of scaffolded enzyme systems on TRAP assemblies.** Relative activity of free enzymes (FDH1 and AlaDH3) and scaffolded enzyme systems (FDH1@TRAP, AlaDH3@TRAP, and FDH1/AlaDH3@TRAP). 100% of AlaDH and FDH activities correspond to 22.1 U·mg$^{-1}$ and 0.83 U·mg$^{-1}$, respectively. The data are presented as the mean of two replicate experiments ($n = 2$). Source data are provided as Source Data file.

subunits of the oligomeric enzymes. The resulting assembly corresponds to a molecular mass of ~370 kDa, consistent with the expected assembly and the 6:3:1 stoichiometry utilized in the incubation (Fig. 1b and Supplementary Table 2). The SEC results confirmed the assembly of the three elements and the changes observed for the hydrodynamic radius (Rh) upon the stepwise enzymatic assembly. These results suggest that TRAP1-3 scaffold can simultaneously assemble both FDH1 and AlaDH3 enzymes in a 3:1 stoichiometry, where the recycling dehydrogenase is excess regarding of the main one.

### Table 1 | Michaelis-Menten steady-state kinetic parameters

| Enzymatic system | Formate | | | |
|---|---|---|---|---|
| | $K_M$ / mM | $V_{max}$ / μmol.min$^{-1}$.mg$^{-1}$ | $k_{cat}$ / s$^{-1}$ | $k_{cat}/K_M$ / M$^{-1}$.s$^{-1}$ |
| FDH1@TRAP | 2.42 ± 1.07 | 0.95 ± 0.20 | 0.7 ± 0.2 | 2.9 × 10$^2$ |
| FDH1/AlaDH3 @TRAP | 6.61 ± 1.71 | 1.34 ± 0.55 | 1.0 ± 0.4 | 1.5 × 10$^2$ |
| Enzymatic system | NAD$^+$ | | | |
| | $K_M$ / mM | $V_{max}$ / μmol.min$^{-1}$.mg$^{-1}$ | $k_{cat}$ / s$^{-1}$ | $k_{cat}/K_M$ / M$^{-1}$.s$^{-1}$ |
| FDH1@TRAP | 0.27 ± 0.10 | 1.23 ± 0.34 | 0.9 ± 0.3 | 3.5 × 10$^3$ |
| FDH1/AlaDH3 @TRAP | 0.34 ± 0.15 | 0.96 ± 0.59 | 0.7 ± 0.5 | 2.1 × 10$^3$ |
| Enzymatic system | Pyruvate | | | |
| | $K_M$ / mM | $V_{max}$ / μmol.min$^{-1}$.mg$^{-1}$ | $k_{cat}$ / s$^{-1}$ | $k_{cat}/K_M$ / M$^{-1}$.s$^{-1}$ |
| AlaDH3@TRAP | 0.39 ± 0.05 | 41.31 ± 2.52 | 30.6 ± 1.9 | 7.9 × 10$^4$ |
| FDH1/AlaDH3 @TRAP | 1.25 ± 0.55 | 37.20 ± 3.34 | 27.6 ± 2.5 | 2.2 × 10$^4$ |
| Enzymatic system | NADH | | | |
| | $K_M$ / mM | $V_{max}$ / μmol.min$^{1}$.mg$^{-1}$ | $k_{cat}$ / s$^{-1}$ | $k_{cat}/K_M$ / M$^{-1}$.s$^{-1}$ |
| AlaDH3@TRAP | 0.36 ± 0.07 | 94.27 ± 6.39 | 69.9 ± 4.7 | 1.9 × 10$^5$ |
| FDH1/AlaDH3 @TRAP | 0.25 ± 0.14 | 71.15 ± 8.01 | 52.7 ± 5.9 | 2.1 × 10$^5$ |

Enzyme kinetic parameters for FDH as FDH1@TRAP and FDH1/AlaDH3@TRAP against the different substrates (formate and NAD$^+$). Enzyme kinetic parameters for AlaDH as AlaDH3@TRAP and FDH1/AlaDH3@TRAP against the different substrates (pyruvate and NADH). $k_{cat}$ were calculated using the MW of the monomers of FDH1 and AlaDH3.

### Catalytic activity of the multi-enzymatic system

The kinetics of free and scaffolded enzymes were evaluated based on UV–Vis assays using different substrates depending on the enzyme activity to be measured. While FDH1 oxidizes formate to $CO_2$ concomitantly reducing NAD$^+$ to NADH, AlaDH3 catalyzes the reduction of that NADH to perform the reductive amination of pyruvate in presence of ammonium chloride to yield L-Alanine as the target product. Both enzyme activities were determined by monitoring either the consumption or the formation of NADH at 340 nm. When FDH1 and AlaDH3 are individually bound to TRAP1-3, they maintained more than 80% activity compared to the activity of their free counterparts (Fig. 2). Furthermore, when both enzymes were assembled into TRAP1-3 scaffold, their specific activities were also unaffected indicating that the scaffolding process keeps the functionality of the assembled enzymes.

To further investigate the catalytic performance of the scaffolded enzymes, we measured the apparent Michaelis-Menten kinetic parameters of the two enzymes either individually scaffolded (FDH1@TRAP and AlaDH3@TRAP) or assembled together (FDH1/AlaDH3@TRAP). As both FDH1 and AlaDH3 are multi-substrate enzymes, we determined the Michaelis-Menten kinetics by varying the concentration of one substrate and fixing saturating concentrations of the others. From the Michaelis-Menten plots (Supplementary Fig. 3), we determined $K_M$, $V_{max}$, $k_{cat}$, and $k_{cat}/K_M$ values for all biocatalysts toward all the substrates (Table 1).

The Michaelis-Menten kinetic parameters clearly show that the rate-limiting step enzyme of this bioredox cascade is the FDH1 as its $k_{cat}$ is two orders of magnitude lower than the $k_{cat}$ of the AlaDH3. The kinetic data evidence that $k_{cat}$ is barely affected by scaffolding the two enzymes together, while $K_M$ values tend to increase when the two enzymes are anchored together within the TRAP1-3 scaffold. The increase in $K_M$ leads to a decrease in the catalytic efficiency ($k_{cat}/K_M$). Remarkably, the increase in $K_M$ was more significant for the smaller substrates such as formate (MW 46) and pyruvate (MW 87) than for the bulkier redox cofactors NAD$^+$ (MW 663) and NADH (MW 664). This increase in $K_M$ indicates that formate and pyruvate have lower affinity for the catalytic centers of the FDH1 and AlaDH3, respectively, when both enzymes are assembled into the TRAP1-3 scaffold. Small substrates such as formate and pyruvate are known to travel toward the enzyme active sites through narrow tunnels embedded in the protein structure[32,33]. In contrast, NAD cofactors are normally bound to dehydrogenases in a more open conformation, inducing a conformational change that facilitates the productive binding of small substrates (aldehydes, ketones, or alcohols). The different access pathways of the two substrates involved in the catalytic mechanism of these two

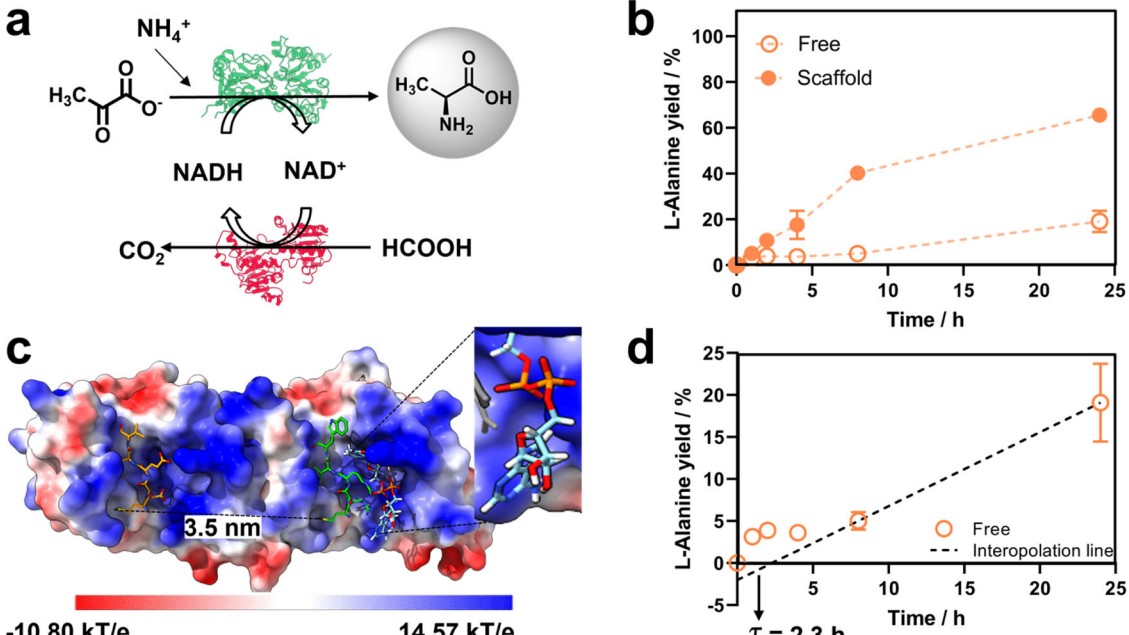

**Fig. 3 | L-Alanine synthesis activity of the scaffolded enzymes in batch-mode.**
**a** Scheme of the catalytic cycle that was tested by the HPLC. **b** L-Alanine yield at 1:1 molar ratio of FDH1:AlaDH3 monomers for the free enzyme system (FDH1/AlaDH3, empty circles) and the scaffolded enzyme system (FDH1/AlaDH3@TRAP, full circles). **c** Structure of the TRAP1-3 scaffold showing the surface electrostatic potential. Peptides 1 and 3 are shown in orange and green sticks, respectively and the distance between the two recognition sites is shown. NADH co-factor docked on the TRAP scaffold is shown in light blue sticks representation. **d** Interpolation line for FDH1/AlaDH3 free system with the transient time highlighted with an arrow. Reaction mixture: 100 mM formate, 75 mM pyruvate, 500 mM ammonium chloride and 0.5 mM NADH. The data are presented as the mean of two replicate experiments ($n = 2$), and error bars represent standard deviations. Source data are provided as Source Data file.

alcohol dehydrogenases may explain why $K_M$ for the small substrates was affected to a higher extent than for NAD cofactors when the enzymes are scaffolded. In the assembled state, the narrower tunnels to allocate either the pyruvate or the formate in the enzyme active sites may be partially compromised by steric impediments that emerged upon the enzyme assembly. Despite the higher $K_M$, the recovered activity upon assembly is still high enough to test this system in a model bioredox cascade. This cascade will transform pyruvate into L-Alanine using ammonium chloride as co-substrate for the reductive amination catalyzed by the AlaDH3 and use formate as the ancillary electron donor for the in situ recycling of NADH, which is orthogonally catalyzed by FDH1.

**Biosynthesis of L-Alanine catalyzed by TRAP-scaffolded enzyme system**

To assess the benefits of the scaffold in the kinetic performance of the multi-enzyme system, we incubated the scaffolded and free biocatalytic systems with pyruvate, 1.3 equivalents of sodium formate, 6.6 equivalents of ammonium chloride and substoichiometric amounts of NADH (150-fold less than pyruvate) (Fig. 3a). Figure 3b shows that the scaffolded system at 63:1 oligomeric molar ratio of TRAP1-3:FDH1:AlaDH3 is five times faster than the free one, reaching and specific productivity of 5.21 g × $g_{enzyme}^{-1}$ × $h^{-1}$ and a titer of 49 mM of L-Alanine after 24 h, which means a chromatographic yield (CY) of 65%. In contrast, free FDH1 and AlaDH3, resulted in a CY of only 19% under the same conditions and incubation time. Considering that the rate-limiting enzyme of this cascade is FDH1 (Table 1), we suggest that scaffolding the system improves the NADH recycling efficiency and thus increases the apparent FDH1 activity. Hence, we hypothesize that the higher specific productivity and product yield achieved with the scaffolded system are the results of a more efficient transport of NAD species between the two proximal enzymes. To further support this hypothesis, we incubated the system with a range of NADH concentrations (0.01–1 mM). We found the maximum performance of the scaffolded system at 0.5 mM NADH (Supplementary Fig. 4). The positive effect provided by the spatial proximity disappears at higher and lower NADH concentrations as the FDH1 may be already saturated at 1 mM NADH in the bulk (three times higher than its $K_M$) or, contrarily, it is too limited by NADH bulk concentrations <0.05 mM (six times lower than its $K_M$). The greatest titer of L-Alanine was also achieved at 0.5 mM, which supports this NADH concentration as the optimal one to maximize the positive effects of the scaffolding on the throughput of the biocascade (Supplementary Table 4). Furthermore, the reaction courses shown in Fig. 3b indicate the existence of a lag in L-Alanine production using the free enzymes. The transient time of the free system was calculated to be 2.3 h by linearly fitting the reaction course with the steepest slope from time 8 to 24 h (Fig. 3d). In contrast, the scaffolded system exhibited no transient time, indicating that the free system takes longer to reach its maximum steady-state production rate compared to the scaffolded one, which reaches maximum throughput at the beginning of the reaction.

To illustrate the spatial proximity of the scaffolded enzymes, we generated a molecular model based on the crystallographic structures of peptides 1 and 3, and TRAP1-3 (Fig. 3c). In this model the predicted distance between assembled FDH1 and AlaDH3 is 3.5 nm. In contrast, at the assay protein concentration (0.18 μM), the free enzymes would be separated by a distance of ~209 nm according to Eq. 1 reported by Ellis et al.[17].

$$d_{e-e} = \frac{1.18}{C^{1/3}} \tag{1}$$

where $d_{e-e}$ is the distance between the free enzymes and C is the concentration of the free enzymes. Considering the inter-enzyme distances ($d_{e-e}$) of both the scaffolded and the free systems, and the hydrodynamic radius of the dimmer FDH1 as rate-limiting enzyme

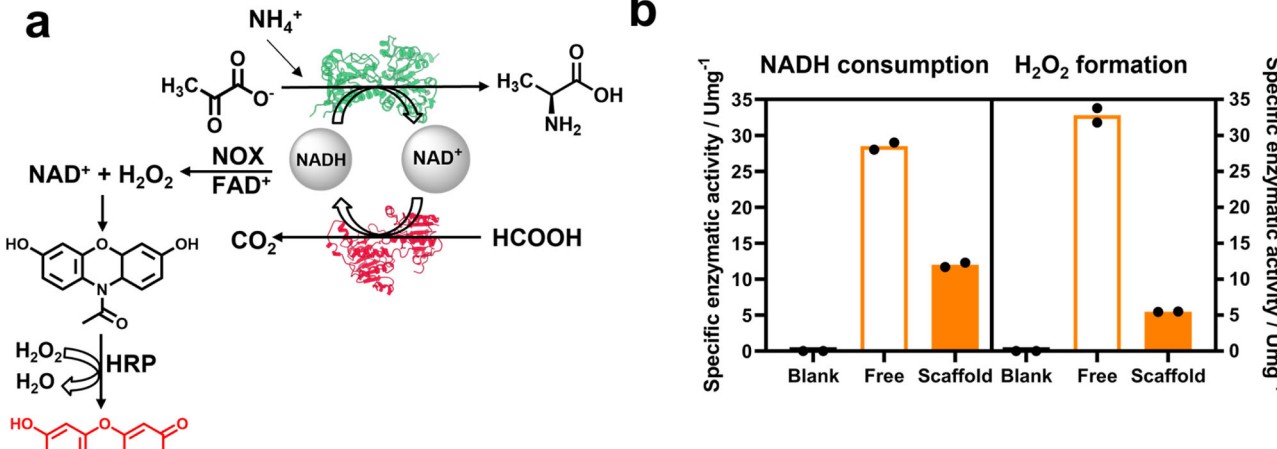

**Fig. 4 | Side reaction competitive catalytic assay. a** Schematic representation of the competitive catalytic assay reaction. **b** Competitive catalytic assay results reported as the consumption of NADH cofactor measured at 340 nm and the formation of $H_2O_2$ by measuring the production of resorufin at 560 nm. Both reactions using separately scaffolded enzymes (FDH1@TRAP/AlaDH3@TRAP, empty bars) or scaffolded enzymes in the same scaffold (FDH1/AlaDH3@TRAP, filled bars) were analyzed at both wavelengths to observe changes in the NADH consumption and $H_2O_2$ formation. Reaction mixture: 100 mM formate, 75 mM pyruvate, 500 mM ammonium chloride, 0.5 mM NADH, 0.15 mM FAD+, 0.1 mg/mL HRP, and 0.05 mM Amplex Red (AR). The data are presented as the mean of two replicate experiments ($n = 2$). Source data are provided as Source Data file.

($r = 2.3$ nm), the fraction of NADH that could be directly channeled from FDH1 to AlaDH3 ($f_{direct}$) is 67% and 1%, respectively, according to the Eq. 2 also presented by Ellis et al.[17]. Thus, the proximity of the two enzymes promoted by the scaffold increases the chances of NAD species to find their next enzyme.

$$f_{direct} = \frac{r}{d_{e-e}} \tag{2}$$

Although the scaffolded system can theoretically channel 67% of NADH formed by the FDH1 to AlaDH3 due to physical proximity (see the previous section), we suggest that the cascade is not directly benefited through channeling by proximity according to Wheeldon et al.[34]. Pure diffusion of NADH from one enzyme to the other is hardly plausible since the diffusion coefficient of this redox cofactor in aqueous media ($4 \times 10^6$ cm$^2$ s$^{-1}$)[35] is much larger than the catalytic efficiency of the rate-limiting enzyme ($3 \times 10^3$ M$^{-1}$ s$^{-1}$); FDH1 bound to TRAP1-3. Instead, we hypothesize that channeling assisted by electrostatic interactions may facilitate the transport of NADH between the two scaffolded dehydrogenases. Surface charge calculations revealed that the surface of TRAP1-3 contains positively charged patches that may interact electrostatically with the negatively charged phosphate groups of the NAD species. Docking studies demonstrate that both reduced and oxidized cofactors can bind to a positively charged cavity within TRAP1-3 (Fig. 3c and Supplementary Fig. 5). This interaction is further supported by an anisotropy-based assay that exploits the intrinsic fluorescence of NADH and results in a TRAP1-3 NADH interaction with a binding constant ($K_D$) of 34.9 μM (Supplementary Fig. 6). Similar electrostatic interactions between the scaffolds and the intermediates have been reported for DNA scaffolds and bi-enzyme systems fused through positively charged linkers[20,21]. Likewise, we suggest that electrostatic interactions between the intermediate (NAD) and the TRAP1-3 scaffolding unit support the channeling we observe.

In the light of these results, the scaffold physically brings the two dehydrogenases together and sequesters NAD cofactors in their surroundings at the concentration herein studied. When the reactions are carried out with 8-fold excess of free FDH1 (1.44 μM) to ameliorate the limitation this enzyme poses to the system, the positive effect of the scaffolding is dramatically reduced (Supplementary Fig. 7). In this scenario the higher FDH1 concentration reduces the average distance between enzymes to 104 nm as calculated with Eq. 1. Using an excess of FDH1 in solution both scaffolded and free systems performed the synthesis of L-Alanine 2 times faster (11.12 g × g$_{enzyme}$$^{-1}$ × h$^{-1}$) and yielded higher product titers (75 mM) in 24 h than using stoichiometric amounts of both dehydrogenases scaffolded in the TRAP1-3. Thus, the positive effect of the scaffolding on the system performance was minimized under these conditions. Overall, these data indicate that NAD species may be channeled between FDH1 and AlaDH3 through a diffusion-limited transport, thus enhancing the mass transfer of the cofactors between the two dehydrogenases.

## Characterization of the channeling effect manifested by the scaffolded multi-enzyme systems

To experimentally support the channeling of NADH between the two dehydrogenases, we performed a competition assay with an NADH-dependent enzyme, and an isotope enrichment assay using deuterated formate. For the competition assay, we performed the reductive amination of pyruvate to L-Alanine in the presence of an excess of free NADH oxidase (NOX) from *Thermus thermophilus HB27,* which oxidizes NADH with the concomitant production of hydrogen peroxide ($H_2O_2$). The $H_2O_2$ generated in situ can be measured in line by adding a fourth enzyme, horseradish peroxidase (HRP), and Amplex red to the assay. If NADH reaches the active site of NOX, this enzyme produces $H_2O_2$ that is further used by HRP to transform Amplex red (AR) into the fluorescent product resorufin (Fig. 4a). Through UV–vis quantification of the produced resorufin, we calculated the $H_2O_2$ generated and consequently, the NADH consumed, by NOX[36,37]. Hence, the more active the NOX is, the less efficient the NADH channeling is between FDH1 and AlaDH3. First, we showed that AlaDH3 and NOX compete for NADH cofactor as NADH is consumed faster and $H_2O_2$ is formed when both enzymes are mixed. The mixture of HRP and AR with the AlaDH3 enzyme assay negligibly generated red color, confirming that $H_2O_2$ could only be formed by NOX. Then, the scaffolded and non-scaffolded systems were incubated with free NOX and HRP in the presence of pyruvate and formate. Figure 4b shows that the NADH consumption was higher with the non-scaffolded system than with the scaffolded one. In agreement, the production of $H_2O_2$ was lower when the enzymes were scaffolded than when they were not. The $H_2O_2$ formation is the consequence of those NADH molecules that derail from the scaffolded cascade and escape to the bulk where are oxidized

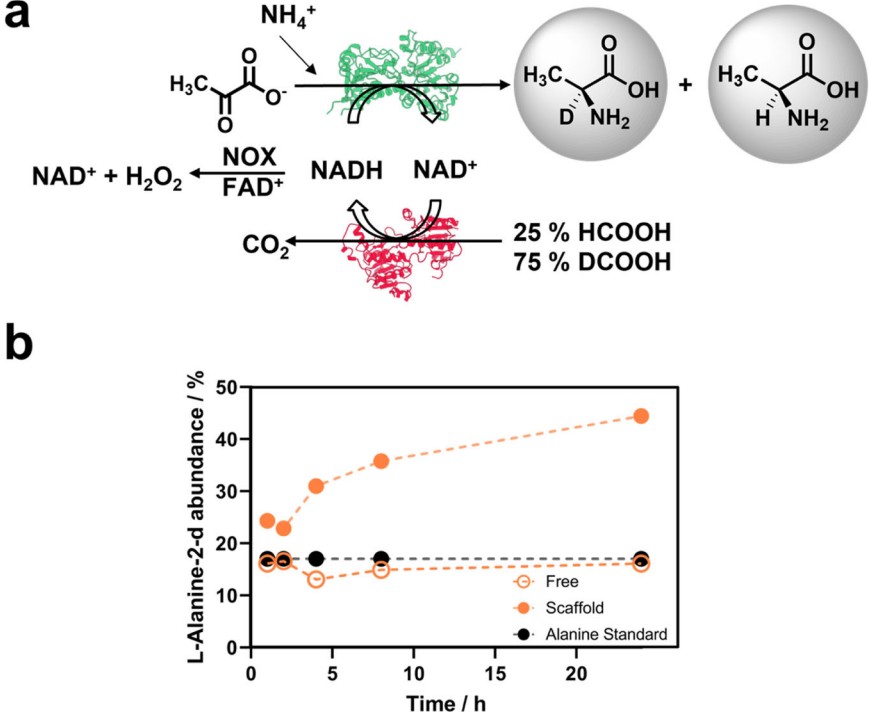

**Fig. 5 | Isotopic enrichment assay coupled to competition assay for the synthesis of L-Alanine-2-d. a** Scheme of the catalytic cycle that was tested by the HPLC-MS. **b** Isotopic abundance of deuterated L-Alanine-2-d ($M/z = 324$) for the free enzyme system (FDH1/AlaDH3, empty circles) and the scaffolded enzyme system (FDH1/AlaDH3@TRAP, filled orange circles). Reaction mixture: 75 mM deuterated formate, 25 mM formate, 75 mM pyruvate, 500 mM ammonium chloride, 0.5 mM NADH and 0.15 mM FAD$^+$. Alanine standard is shown in filled black circles. Source data are provided as Source Data file.

by NOX. Likewise, higher consumption of NADH means that FDH1 is not efficiently replenishing the pool of this reduced cofactor. Therefore, when FDH1 and AlaDH3 are scaffolded, NOX competes with AlaDH3 to a lower extent than when they are separated and diluted in the bulk. The competition assay informs us about the restricted leakage of NADH from the microenvironment of the scaffolded enzymes, supporting the fact that the cofactor regeneration, and thus the overall performance of the cascade, are more efficient when the enzymes co-localize in the TRAP-scaffold.

To confirm the channeling of NADH between the two dehydrogenases, we performed a more complex experiment where an isotopic enrichment assay was merged with the competition assay previously mentioned. To do so, we performed the reaction under the presence of deuterated formate (see experimental section). In this assay, FDH1 produces deuterated NADH that is concurrently used by the neighbor AlaDH3 when they are scaffolded yielding deuterated L-Alanine-2-d. If the system is not scaffolded the deuterated NADH is released by FDH1 to the media where NOX can oxidize it back to NAD$^+$ (Fig. 5a). Under these conditions deuterated NADH could transfer the deuterium to pyruvate only if NADH is directly channeled from FDH1 to AlaDH3 (see methods). Unfortunately, the deuterated formate completely inhibited the FDH1 activity (Supplementary Fig. 8), thus the experiment was performed with a mixture of deuterated and non-deuterated formate (75 mM and 25 mM). After 24 h, 40% of the L-Alanine synthesized was deuterated for the scaffolded system, whereas deuterated L-Alanine was undetected for the non-scaffolded system (Fig. 5b). This enrichment in the heavy isotope of L-Alanine demonstrates that as soon as NADH is deuterated by FDH1 assembled on TRAP1-3, the neighboring enzyme, AlaDH3, uses this channeled cofactor to synthesize the deuterated L-Alanine. This experiment confirms the channeling of the cofactor between the two scaffolded dehydrogenases through a diffusion limited effect created by the postulated electrostatic interactions between the nicotinamide cofactors and the TRAP scaffold. This cofactor channeling

contributes to speed up the cascade reaction and increases the final product yield.

## Heterogenization of the enzyme scaffolds

Once proved that the scaffolded system enhances the production rate and yield of the biosynthesis of L-Alanine through scaffold-assisted channeling of NADH, we intended to assemble the multi-enzyme system in solid-phase. Unfortunately, the solid-phase assembly failed to assemble the system with the optimal oligomeric stoichiometry 6:3:1 (Supplementary Table 5 and Supplementary Fig. 9). To guarantee these ideal TRAP1-3:FDH1:AlaDH3 stoichiometry in the solid surface of the support, we first assembled the scaffold in solution and then immobilized it on porous agarose beads Herein, the optimal stoichiometry of the scaffold in solution assures that both enzymes are close each other even when they are immobilized. As carrier, we exploited a tri-functional agarose-based carrier recently developed, which displays cobalt chelates, positively charged amine groups, and aldehydes at its surface[38]. The cobalt chelates drive the immobilization of the enzyme assembly through the His-tag fused to the TRAP1-3 unit, while aldehydes and amines establish covalent and ionic bonds with the enzyme complexes, respectively (Supplementary Fig. 10). The load of FDH1/AlaDH3@TRAP renders enzyme loads of 2.8 mg × g$_{carrier}^{-1}$ for AlaDH3 and FDH1 (Supplementary Table 5). CLSM analysis reveal that the two enzymes are co-immobilized, but also co-localize on the same particle (Fig. 6a–c) as their radial profiles and relative infiltration distances across the porous structure of the beads perfectly match (Fig. 6d, e). Furthermore, the colocalization analysis support that both enzymes greatly colocalize (Person coefficient of 0.85 ± 0.04 and Manders coefficients >0.99) (Supplementary Table 6). Moreover, the index of correlation obtained through colocalization map analyses presented similar conclusions (0.58 ± 0.04)[39,40]. Figure 6f shows higher values of the normalized mean deviation product (nMDP) in the outer surface (yellow pixels) and in the large voids of the agarose beads, pointing out that the fully assembled scaffold is mainly localized in

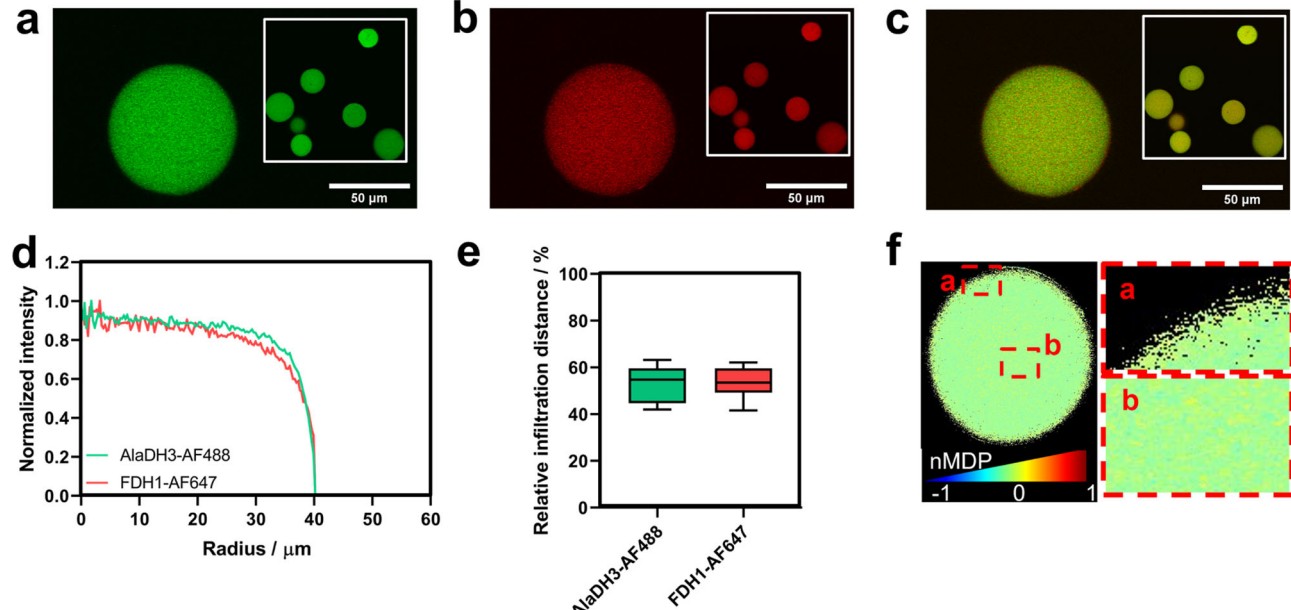

**Fig. 6 | Confocal fluorescence microscopy images (20X magnification) of co-immobilized FDH1/AlaDH3@TRAP on tri-functional carrier.** AlaDH3 and FDH1 were labeled with Alexa Fluor-488 (AF488) and Alexa-Fluor 647 (AF647). **a** Spatial distribution of AlaDH3 (green channel), **b** FDH1 (red channel) and **c** overlay of the two fluorophores. The inlet shows micrographs at lower magnification. **d** Radial profile and **e** relative infiltration distance of labeled AlaDH3 (green line and bar) and FDH1 (red line and bar). Relative infiltration distance is defined as the fraction of the radius where the fluorescence intensity was higher than 50% of the maximum

intensity. The data are presented as the mean of ten replicate experiments ($n = 10$), and error bars represent standard deviations. **f** Co-localization map created with Colormap Image J plugin. The scale refers to the co-localization degree of the two labeled enzymes. Blue pixels (value = −1) mean the absence of co-localization. Red pixels (value = 1) mean a high degree of co-localization. The regions framed with the dashed red square represent those pixels with co-localization values > 0.1 where the spatial colocalization of the two fluorophores is statistically significant. Source data are provided as Source Data file.

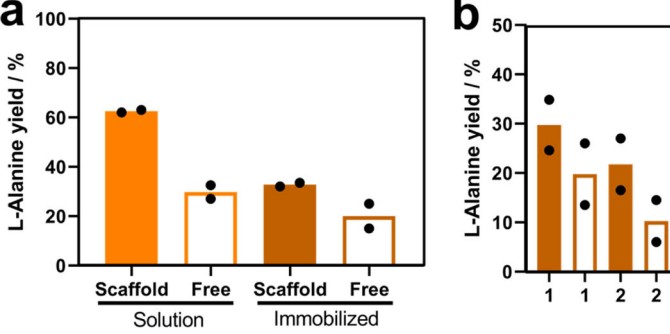

**Fig. 7 | Operational performance and reusability of the scaffolded enzyme systems immobilized on porous carriers. a** L-Alanine yield % comparing different multi-enzyme systems in solution and immobilized after 24 h. **b** L-Alanine yield % in a reusability test for free (FDH1/AlaDH3) (empty bars) and scaffolded (FDH1/

AlaDH3@TRAP) (orange bars) enzyme systems. Reaction mixture: 100 mM formate, 75 mM pyruvate, 500 mM ammonium chloride and 0.5 mM NADH. The data are presented as the mean of two replicate experiments ($n = 2$). Source data are provided as Source Data file.

highly open porous regions where steric hindrances are minimized. In parallel, we co-immobilized FDH1 and AlaDH3 without His-tag on the same carrier to fabricate a benchmarked heterogeneous biocatalyst that allowed us studying the effect of the spatial enzyme arrangement in the confined space. In this latter case, the two enzymes were randomly immobilized through electrostatic and covalent interactions established between acidic residues (Asp and Glu) and Lys of the enzymes and the amine and aldehyde groups of the carrier, respectively. The enzyme load was similar to the scaffolded system (3.3 mg × $g_{carrier}^{-1}$ for both enzymes, Supplementary Table 5), maintaining the same AlaDH3:FDH1 ratio. In this case, the two enzymes co-localized to a lower extent than when the scaffolded system is immobilized but to a larger extent than when the scaffold is assembled in solid phase (Supplementary Fig. 11 and Supplementary Table 6).

Once the scaffolded and non-scaffolded systems were prepared and characterized at the sub-micrometric level, they were challenged for the reductive amination of pyruvate with in situ NADH recycling using formic acid as ancillary electron donor[41]. We assessed the performance of the different spatial arrangements both in solution and immobilized. In all cases, we adjusted the monomeric molar ratio of the four systems to 1:1 FDH1:AlaDH3. Reactions were carried out as described above and the L-Alanine yield was determined at 24 h by HPLC. Figure 7a shows that L-Alanine yield decreases two times when the scaffolded system was immobilized compared to its counterpart in solution. Likewise, the specific productivity of the scaffolded system is solution was 5 times higher than its immobilized counterpart (Supplementary Table 7). The lower yield and productivity measured upon immobilization might be explained by both external and internal

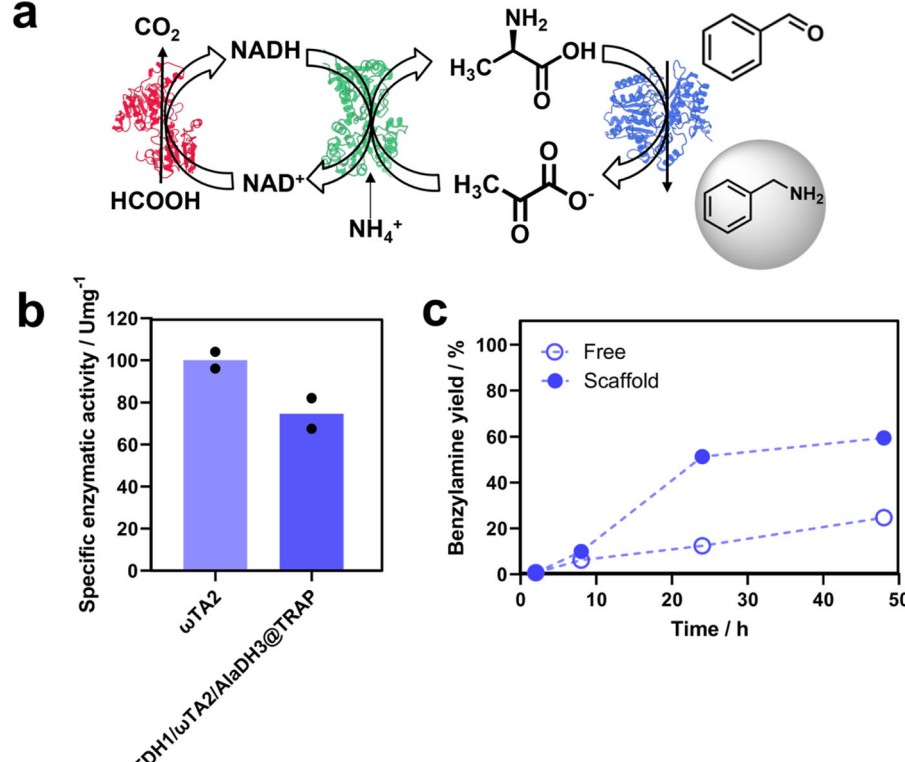

**Fig. 8 | Benzylamine biosynthesis using a three-enzyme organized multi-enzymatic system. a** Scheme of the parallel reactions to produce benzylamine. **b** Relative activity of ωTA2 in the free and scaffolded form (FDH1/ωTA2/AlaDH3@TRAP). 100% of ωTA activity corresponds to 5.58 U·mg⁻¹. **c** Benzylamine chromatographic yield % at an enzyme monomer ratio of 1:1:1 for in free (FDH1/ ωTA2/AlaDH3) and scaffolded system (FDH1/ωTA2/AlaDH3@TRAP). Reaction mixture: 10 mM benzaldehyde, 50 mM pyruvate, 500 mM ammonium formate, 0.1 mM PLP and 0.5 mM NADH. The data are presented as the mean of two replicate experiments ($n = 2$). Source data are provided as Source Data file.

diffusion limitations typically observed for heterogeneous biocatalysts. To note, the scaffolded multi-enzyme system achieved roughly two times larger L-Alanine chromatographic yields regardless of whether it was immobilized or free. Then we recycled the two scaffolded and non-scaffolded immobilized systems for consecutive operational cycles. Upon each 24 h reaction cycle, the solid biocatalysts were separated from the reaction media through vacuum filtration and washed before starting a new cycle. The reaction crudes were analyzed by HPLC to determine the L-Alanine yield. Figure 7b shows a decay of the product yield along the cycles indicating the inactivation of the heterogeneous biocatalysts. The immobilized and scaffolded system was more stable than its non-scaffolded counterpart as the former reached 18% product yield after the third cycle, compared to the 10% yield achieved by the latter. Despite none of the tested immobilized systems achieved 100% product yield and were relatively unstable during their operational use under the tested conditions, the scaffolding of the two enzymes overall enhances the productivity and the operational stability of the multi-enzyme system. To understand if the biocatalysts inactivation was due to either enzyme leakage or intrinsic enzyme inactivation, we analyzed the reaction crudes and the washes by SDS-PAGE. Since both immobilization chemistries for scaffolded and non-scaffolded systems rely on a covalent immobilization chemistry, leaching of the working enzymes should only occur if some enzyme subunits do not directly interact with the carrier surface. SDS-PAGE analysis negligibly detected proteins in the reaction crude and wash solution after each cycle when the enzyme system was scaffolded and immobilized, on the contrary when the two enzymes were co-immobilized but not scaffolded, we can detect the lixiviation of some AlaDH subunits (Supplementary Fig. 12) after two reaction cycles. Therefore, we suggest that the decay in the performance of the co-immobilized but not scaffolded system is practically due to the dissemble of the quaternary structure of AlaDH, which is

stabilized to a higher extend when it is scaffolded. Nonetheless, we cannot discard that beside lixiviation, the activity loss of both systems is also driven by inactivating structural distortions that occur during their operational use. Therefore, product yield and SDS-analysis support the fact that the scaffolding of the system promotes a significant overall operational stabilization of the assembled multi-enzyme system.

## Amine biosynthesis catalyzed by a TRAP-scaffolded tri-enzyme system

Encouraged by the enhancement that the TRAP-based scaffold promoted in the redox FDH1/AlaDH3 system for the synthesis of L-Alanine, we went one step further and assembled a tri-enzyme system onto the TRAP scaffold[25]. As third enzyme we selected the ω-transaminase from *Pseudomonas fluorescens* (ωTA). The aim of incorporating this new enzyme was to construct a cascade with three orthogonal reactions to in situ recycle both redox cofactors and amine donors. This cascade has been previously designed to aminate benzaldehyde to benzylamine using L-Alanine as amine donor (Fig. 8a)[42]. This amine donor is in situ regenerated from the pyruvate released by the ωTA, ammonium formate and NADH through a reductive amination catalyzed by AlaDH. Finally, the NADH pool is replenished by the action of FDH which uses formate as hydride donor to reduce the NAD⁺ to NADH (Fig. 8a). Thus, the couple AlaDH3/FDH1 allows the recycling of both the amine donor and the redox cofactor[42,43]. This cascade is thermodynamically challenging as concentration-based equilibrium constant of the amine donor (L-Alanine) and the molar ratio between the amine donor and amine acceptor (benzyladehyde) are extremely low from the beginning of the reaction[44,45].

To assemble this tri-enzyme system we constructed a new TRAP-scaffold with an additional binding domain (TRAP2) for anchoring ωTA. TRAP2 domain has high affinity ($K_D = 1.7\,\mu M$) and selectivity

toward the peptide-2 MERVW, and is orthogonal to peptide-1 and peptide-3 displaying weaker affinity with $K_D$ values of 42.1 μM and 83 μM, respectively[25]. In parallel, this peptide-2 was fused to the C-terminus of ωTA giving rise to the variant ωTA2. Before carrying out the assembly, the structural integrity and functionality of ωTA2 and TRAP2-3-1 were characterized by SDS-PAGE, MALDI-TOF and CD, confirming the expected size and secondary structure according to their primary sequences (Supplementary Table 1). The new trivalent scaffold (TRAP2-3−1) and the corresponding tagged enzymes were assembled following the stepwise protocol described above. The assembly sequence was AlaDH3, FDH1, and ωTA2 to achieve the tri-enzyme scaffold. SEC analysis showed a main peak that corresponded to a molecular size of 605 kDa, which indicates that the scaffold was also assembled in a equimolar ratio 1:1:1:1 of TRAP2-3−1:FDH1:ω-TA2:AlaDH3 of the monomeric units, according to the primary sequence of all elements. In addition, when this SEC peak was analyzed by SDS-PAGE gel electrophoresis, we could identify the three enzymes and the scaffold (Supplementary Fig. 2c). The specific enzyme activity of the ωTA enzyme was significantly affected after fusion to peptide-2 with a 42% decrease in activity (Supplementary Table 3).

After assembly, the activity of ωTA2 was spectrophotometrically analyzed by the broadly used transaminase assay based on the dea-mination of methylbenzylamine[46]. Figure 8b shows that the scaffolded ωTA2 exhibited 74% of the activity of its free counterpart. Similarly, to FDH1 and AlaDH3, the reduction of the enzyme activity upon the assembly might be owing to steric hindrances the small substrates (L-Alanine and benzaldehyde) suffer to reach the more crowded envi-ronment of scaffolded ωTA2. When the scaffold FDH1/ωTA2/AlaDH3 was incubated with benzaldehyde, an excess of pyruvate and ammo-nium, and substoichiometric amounts of NADH, we observed that the benzylamine chromatographic yield upon 24 h was 50% when using the scaffolded system; a yield 4.25-fold larger than using the free sys-tem (Fig. 8c). While the scaffold system reached a plateau after 24 h achieving a titer of 5.9 mM benzylamine, the free system keeps work-ing until reaching a titer of 1.4 mM benzylamine after 48 h. Likewise, the system productivity was enhanced 2.36 times when the three enzymes were scaffolded compared to the free system after 48 h. The maximum specific productivity of the scaffolded biocatalyst was 0.047 g × $g_{enzyme}^{-1}$ × $h^{-1}$, a value that is 8-times larger than the reductive aminating cascade composed by ωTA from *Aspergillus terreus*, AlaDH, and FDH reported elsewhere[47]. The scaffolded system reaches the same specific productivity as ωTA cross-linked enzyme aggregates (CLEA) working as sole enzyme in the reductive amination of benzal-dehyde using a 10 times molar excess of L-Alanine as amine donor (0.049 g × $g_{enzyme}^{-1}$ × $h^{-1}$)[30]. Despite the formation of benzylamine in presence of an excess of pyruvate is thermodynamically unfavoured, the physical proximity of the amine donor generating enzymes (FDH and AlaDH) and the ωTA afford an amination reaction otherwise lim-ited using a diluted free enzyme system. This scaffolded system with three enzymes is one of the few examples of assemblies with more than two enzymes using biomolecular scaffolds that do not rely on enzyme clustering. TRAP2-3-1 is able to form a supramolecular complex that gathers two dehydrogenases and one transaminase to force an inter-enzyme channeling from one enzyme to its assembled neighbor in a thermodynamically unfavored cascade. As well as in the case of the bi-enzyme system FDH1/AlaDH3, we suggest a channeling driven by NADH-TRAP2-3-1 interactions.

## Discussion

In conclusion, a simple methodology has been developed for assem-bling of enzymes onto a scaffold based on engineered Tetratricopep-tide Repeat Affinity Proteins (TRAP). These modules can be coupled to form arrays of specific and orthogonal recognition protein domains that selectively bind short peptide sequences fused to enzymes. This is a methodology driven by a generally applicable biomolecular

recognition which serves as a tool for preparing scaffolded multi-enzyme systems. In addition to the spatial organization achieved by this strategy, it should be note that TRAP scaffolds can also be engi-neered to electrostatically interact with cofactors and reaction inter-mediates, providing a second level of tunability to the system. To the best of our knowledge, this is the first example of a protein-scaffold designed to organize several enzymes at the nanoscale and capture reaction intermediates to increase their local concentration in the surroundings of the scaffolded enzymes. This has been shown pre-viously with DNA scaffold but never with protein-based ones.

These concepts have been proven with a two-enzyme system that performs the simultaneous asymmetric reduction of α-ketoacids to L-amino acids while recycling NADH. These enzymes were suc-cessfully assembled on the TRAP1-3 scaffold via biomolecular recognition with a controlled stoichiometry, where the largest enzyme acts as a nucleation point for the assembly. The efficiency of the biocatalytic cascade was significantly improved upon assembly due to a diffusion-limited strategy driven by electrostatic cofactor-scaffold interactions, resulting in the effective channeling of NADH from FDH1 to the adjacent AlaDH3 and boosting in situ NADH recy-cling. This channeling effect, albeit to a lesser extent, also occurred when the enzyme scaffold was immobilized on porous beads. Finally, a more complex system composed of three-tagged enzymes FDH1, ωTA2, and AlaDH3, was assembled on a scaffoldin unit with three orthogonal TRAP domains (TRAP2-3-1). This cascade with three par-allel reactions allows in situ recycling of both redox cofactors and amine donors. The assembly of this three-enzyme cascade demon-strated not only that the methodology developed allows the cou-pling of three step cascade reactions, but also that the throughput for the cascade is enhanced by arranging the enzymes at the nanoscale and by increasing the local concentration of the cofactors by directed interactions with the scaffold.

The methodology developed here is relatively simple and mod-ular compared to other current approaches. This is due the following facts: (1) the scaffolds are exquisitely orthogonal; (2) the peptide tag-ging approach is simple, as only short 10−20 amino acid sequences are needed to anchor enzymes to the scaffolding; (3) the approach is easily genetically programmed, as we constructed a set of plasmids for block cloning and tagging of any enzyme; (4) the peptide tagging has not resulted in any major defect on the protein activity for all the systems tested to date; (5) the cofactor interactions are driven by easily encodable electrostatic interactions; and finally (6) the methodology also involves a straightforward incubation process for assembling the components, which makes it relatively simple and scalable for use in large-scale applications.

We envision that this technology will make a strong contribution to advancing the manufacture of more robust multi-enzyme systems, where enzymes are scaffolded and organized with nanometric preci-sion. Our endeavor has been based on creating a relevant multi-enzyme system that allows for the assembling of enzymes capable of acting as biocatalysts on protein-based scaffolds. Therefore, the combination of multifactor protein engineering and biocatalysis exhibits enormous potential not only to enhance the intrinsic catalytic activity and stability of enzymes but also to maximize the performance of spatially orga-nized multi-enzyme systems[48]. The controlled interactions with key cofactors and reaction intermediates make this technology even more promising for a wide range of applications in biocatalysis. Furthermore, the applications demonstrated in this work could be expanded to other fields of applied science, for instance, the integration into energy devices[49], or the formation of biocatalytic films[50].

## Methods

### Protein design, cloning and molecular biology

TRAP1-3 and TRAP2-3-1 scaffolds are a combination of two and three Tetratricopeptide Repeat Affinity Proteins, each TRAP represents an

engineered TPR module that binds to its cognate peptides, MEEVV, MERVW, and MRRVW for TRAP1, TRAP2, and TRAP3, respectively[25]. TRAP1-3 and TRAP2-3-1 genes were purchased from Biomatik, USA. These genes were cloned in the pet-28b (+) vector, which has been selected as the standard vector for the expression of the proteins. The gene was ordered flanked with four restriction sites for cloning in pet-28b (+) or in pProEx-HTa vector, a plasmid commonly used to express TPR proteins. The enzymes FDH, ωTA, and AlaDH cloned in pet-28b (+) plasmid were tagged with the two cognate peptides for TRAP1, TRAP2, and TRAP3 proteins (MEEVV in the case of FDH enzyme, MERVW in the case of ωTA, and MRRVW in the case of AlaDH enzyme) by overlapping PCR. The tagged enzymes amplified fragments were cloned into pet-28b (+).

## Protein expression and purification
TRAP proteins and tagged enzymes (enzymes fused to peptides) were overexpressed in *Escherichia coli* C41 cells. An overnight saturated cell culture was diluted in 1 L of LB and grown to an OD600 = 0.6–0.7 at 37 °C. The next step was to induce over-expression by the addition 0.6 mM IPTG follow by overnight growth at 20 °C. Proteins were purified as His-tagged fusions following standard protocols using nickel nitrilotriacetic acid affinity chromatography (Ni-NTA)[51]. Protein concentration was estimated based on the extinction coefficients calculated from their amino acid composition and the absorbance at 280 nm. TRAP proteins were stored at −80 °C. FDH1 and AlaDH3 tagged enzymes were stored with 40% glycerol at −80 °C, and ωTA2 was stored with 10% DMSO at −20 °C to preserve their activity.

## MALDI-TOF mass spectrometry
MALDI-TOF measurements were performed using the Voyager-DE PRO Biospectrometry Workstation mass spectrometer. The samples were prepared in PB buffer (Phosphate buffer 10 mM pH 7.4). 1 μL of protein was combined with 4 μL of matrix. The preparation of the matrix solution was carried out by mixing 50:50 of acetonitrile-water and 0.1% of TFA (Trifluoroacetic acid) with sinapinic acid matrix at 10 mg·mL⁻¹ final concentration[52].

## Circular dichroism
Circular dichroism (CD)[53,54] experiments were performed in PBS buffer (10 mM NaCl, 10 mM phosphate buffer pH 7.4) at 10 μM protein concentration using a Jasco J-815 spectrophotometer. Protein spectra were measured from 190–260 nm. In addition, thermal denaturation curves were acquired in the range of temperatures between 15 and 100 °C by recording data every 1 degree.

## Enzyme assembly onto TRAPs proteins
The assembly between TRAP scaffold (TRAP1-3) and tagged enzymes (FDH1 and AlaDH3) is promoted by specific biomolecular recognition interaction[25]. Before incubation, the His-tag of the tagged enzymes was removed by cleavage with TEV protease, whereas the His-tag of TRAP1-3 was not removed. The assembly was carried out in a sequential manner. Initially, AlaDH3 hexamers were incubated 1 h at 4 °C with TRAP1-3 monomers at a 1:6 stoichiometry to load all the AlaDH3 monomers with one TRAP unit. Then, FDH1 dimers were incubated for 1 h at 4 °C at a stoichiometry of one dimer per two TRAPs, resulting in an assembly with a final stoichiometry of 6:3:1 (TRAP1-3:FDH1:AlaDH3), i.e., 1:1:1, when considering the monomeric units. Finally, the assemblies were purified by FPLC gel filtration through a Superdex 200 (10/300) column using an ÄKTA Pure protein purification system. The chromatography was carried out in PBS buffer (150 mM NaCl, 50 mM phosphate buffer pH 7.4) at a flow rate of 0.5 mL/min, and the chromatogram recorded at 280 nm.

The assembly between the TRAP scaffold (TRAP2-3-1) and the three tagged enzymes (FDH1, ωTA2, and AlaDH-3) was promoted by the specific biomolecular recognition interaction as the two-enzyme assembly. The incubation time, temperature, and molar ratio between TRAPs and tagged enzymes was similar to the two enzyme systems with an stoichiometry of 6:3:1,5:1 for TRAP2-3-1:FDH1:ωTA2:AlaDH3, when considering the oligomeric units (i.e. 1:1:1:1 for the monomeric units). Finally, the assembly was purified by FPLC gel filtration through a Superdex 200 (10/300) column under the same conditions described above for the two-enzyme scaffolded system.

## Determination of the hydrodynamic radius (Rh)
The Rh of the TRAP scaffold, free enzymes, and assembled enzymes systems was measured by microfluidic diffusional sizing technology using the Fluidity One W system[31]. The system employs a unique diffusion-based technique to calculate size-related metrics of fluorescently labeled molecules. TRAP1-3 scaffold was labeled with Alexa-Fluor 647 (AF647) as fluorescent reporter to determine the diffusion and thus the Rh. Proteins were incubated for 1 h with A647 dye solution in DMSO at 1:3 protein:dye molar ratio, in darkness with agitation at 25 °C. The excess of the dye was removed by gel filtration through a PD-10 column. The Rh of the free tagged enzymes and TRAP1-3 scaffold was analyzed at 2 μM protein concentration. To determine the Rh of the assemblies unlabeled tagged enzymes were incubated sequentially with labeled TRAP1-3 scaffold. First, AlaDH3 and then FDH1 as described in the previous methods section at a 6:3:1 ratio for TRAP1-3:FDH1:AlaDH3, monomer, dimer and hexamer, respectively. The Rh of the different scaffolded enzyme systems was measured by tracking the fluorescence of the TRAP1-3 scaffold at 2 μM concentration.

## Fluorescence anisotropy-based binding assay
The binding of NADH cofactor to TRAP1-3 scaffold was conducted in PBS buffer (150 mM NaCl, 50 mM PB pH: 7.4) in a 10 × 10 mm path-length cuvette at 25 °C at 1 μM NADH concentration. After 5 min of equilibration, increasing quantities of the TRAP1-3 scaffold (from 0 to 100 μM TRAP1-3 concentration) were added to the NADH solution (100 μL at 1 μM NADH concentration), and the fluorescence anisotropy was measured. Experiments on fluorescence anisotropy were carried out using a Fluorometer NIR fluorescence spectrophotometer outfitted with excitation and emission polarizers. Fluorescence anisotropy was measured with excitation at 340 nm and emission at 463 nm using slits of 6 nm.

Anisotropy was calculated using the Eq. 3 G-factor: adjustments were performed to account for the difference in transmission efficiency of the two emission channels.

$$G - factor = \frac{I_{HV} - I_{B,HV}}{I_{HH} - I_{B,HH}} \tag{3}$$

where $I_{HV}$ is the vertical emission (0°) of a standard solution with excitation in horizontal orientation (90°), $I_{HH}$ is the horizontal emission of a standard solution with excitation in horizontal orientation, $I_{B,HV}$ is the vertical emission of a blank solution with excitation in horizontal orientation, and $I_{B,HH}$ is the horizontal emission of a blank solution with excitation in horizontal orientation. PBS buffer (150 mM NaCl, 50 mM PB pH: 7.4) was used as a blank solution and 1 μM NADH as a standard solution.

The Eq. 4 for anisotropy ($r$) includes the G-factor for excitation at vertical orientation (0°) is:

$$r = \frac{(I_{VV} - I_{B,VV}) - G(I_{VH} - I_{B,VH})}{(I_{VV} - I_{B,VV}) + 2G(I_{VH} - I_{B,VH})} \tag{4}$$

where G is the G-factor, $I_{VV}$ and $I_{VH}$ are the sample's vertical and horizontal emission, respectively, and $I_{B,VV}$ and $I_{B,VH}$ are the intensity of emission of the blank with the emission polarizer in vertical and horizontal orientations, respectively.

To calculate the percentage of NADH bound at different concentrations it was used the Eq. 5[55]:

$$Binding\% = \frac{r - r_f}{r_b - r_f} \tag{5}$$

where $r$ represents the measured anisotropy for NADH at any TRAP1-3 scaffold concentration, $r_f$ represents the anisotropy of free NADH, and $r_b$ represents the anisotropy of NADH bound to the TRAP1-3 scaffold in the plateau area of the binding curve. GraphPad Prism 9 software was used to fit the data to a One Site-Specific binding model.

## Enzymatic activity measurements

The activity of the free enzymes, tagged enzymes and scaffolded enzymes was measured in solution[56]. To determine the AlaDH3 activity, the consumption of NADH was measured by the decrease in absorbance at 340 nm under the following reaction conditions: 0.5 mM NADH, 75 mM pyruvate, 500 mM ammonium chloride in 25 mM potassium phosphate pH 8.0 at 0.2 μM AlaDH3. To determine the FDH1 activity, the production of NADH was measured by the increase in absorbance at 340 nm under the following reaction conditions: 1 mM NAD+, 100 mM sodium formate, in 25 mM sodium phosphate buffer pH 7.0 at 2 μM FDH1. All the measurements were performed in 200 μL reaction volume during 30 min, at a 340 nm wavelength and at 30 °C using Synergy H1 Hybrid Multi-Mode Microplate Reader from BioTeK Instrument in 96-well UV–Vis transparent plates. To determine the activity of the ωTA2 enzyme, the conversion of methylbenzylamine (FEA) to acetophenone was measured by monitoring an increase in the absorbance at 245 nm under the following reaction conditions: 2 mM pyruvate, 0.1 mM PLP (Pyridoxal 5′-phospate monohydrate) and 2 mM FEA in acetonitrile in 200 mM HEPES buffer pH 8.0 at 0.5 μM ωTA2. The slope of absorbance as a function of time was calculated by a linear fit of the first time points of each reaction to calculate the enzyme activity units according to the Lamber-beer equation. For FDH1 and AlaDH3, the NADH concentration was calculate using the extinction coefficient (ε) $\varepsilon_{NADH\text{-}340\ nm} = 6200\ M^{-1} \times cm^{-1}$. One unit of FDH1 is defined as the amount of enzyme needed to produce 1 μmol of NADH per minute under the above given conditions. One unit of AlaDH3 is defined as the amount of enzyme needed to consume 1 μmol of NADH per minute under the above given conditions. For the ωTA2, we used the extinction coefficient (ε) of acetophenone; $\varepsilon_{acetophenone\text{-}245\ nm} = 12,000\ M^{-1} \times cm^{-1}$. One unit of ωTA2 is defined as the amount of enzyme needed to produce 1 μmol of acetophenone per minute under the above given conditions. The specific activity was calculated normalizing the activity units per the enzyme concentration in the assay (U·mg⁻¹).

## Enzyme kinetic parameters

Kinetic parameters: Michaelis constant ($K_M$), maximum rate ($V_{max}$), turnover number, ($k_{cat}$) and catalytic efficiency ($k_{cat}/K_M$) of tagged enzymes (FDH1 and AlaDH3) were determined spectrophotometrically by Synergy H1 Hybrid Multi-Mode Microplate Reader from BioTeK Instrument in 96-well UV–Vis transparent plates according to the activity assays described above. Different concentration ranges of the cofactors NADH (from 0 to 0.66 mM) and NAD+ (from 0 to 8.33 mM), pyruvate (from 0.9 to 200 mM), formate (from 0 to 62.5 mM), and ammonium chloride (500 mM) were evaluated to calculate the $K_M$ and $V_{max}$. Experimental data were fitted by the Michaelis-Menten Eq. 6[57]:

$$V = \left( \frac{V_{max} \times [S]}{K_M + [S]} \right) \tag{6}$$

where V is the initial velocity, [S] is the substrate concentration, $V_{max}$ the maximum reaction velocity, and $K_M$ the Michaelis-Menten constant. The activity of AlaDH3, in which pyruvate is the substrate, since the data does not fit well to a simple Michaelis Menten equation it

was fitted using the following substrate inhibition Eq. 7[58]:

$$V = V_{max} \frac{[S]}{[S] + K_M \left(1 + \frac{I}{Ki}\right)} \tag{7}$$

where V is the initial velocity, $V_{max}$ is the maximum velocity, $K_M$ is the substrate binding constant (Michaelis-Menten substrate affinity constant), and $K_i$ the inhibitor binding constant.

The errors reported for the kinetic parameters, $K_M$ and $V_{max}$, were calculated from the average of three replicates. The turnover numbers ($k_{cat}$) were obtained using the Eq. 8:

$$k_{cat} = V_{max} / [\text{Enzyme}] \tag{8}$$

The ratio $k_{cat}/K_M$ defines the catalytic efficiency of the systems.

## Biotransformation of L-Alanine

L-Alanine was synthesized in batch incubating the different biocatalysts AlaDH3 and FDH1 at two different AlaDH3:FDH1 molar ratios 1:1 and 1:8, which correspond to of 0.28:0.28 μM, and 0.28:2.25 μM AlaDH3:FDH1 concentrations in 1 mL of the reaction mixture composed by 75 mM pyruvate, 100 mM sodium formate, 500 mM ammonium chloride and 0.5 mM NADH in nano pure water. Reactions were incubated under orbital agitation at 500 rpm and 25 °C for 1, 2, 4, 8, and 24 h. The reaction was stopped, and the L-Alanine was collected at each time point by passing the samples through an ultrafiltration unit Amicon Ultra-0.5 Centrifugal Filter Units, which were centrifuged 30 min at 13100 rcf. The conversion degree of L-Alanine was confirmed by chiral derivatization with Marfey's reagent[59] and analyzed by HPLC. Briefly, 20 μL of 1:10 diluted reaction samples were mixed with 8 μL of 1 M sodium bicarbonate and 20 μL of 15 mM Marfey's reagent (Cat. 48895, Thermo Scientific) in acetone and incubated for 1 h at 50 °C and 400 rpm. Then, the derivatized reaction was stopped by the addition of 8 μL of 2 M HCl and then centrifuged at 6700 rcf for 15 min. In addition, the supernatant was filtered to perform the HPLC analysis. Derivatized samples were analyzed in a HPLC Agilent Technologies 1120 Compact LC, with an EC-C18 2.7 μm column (4.6 × 100 mm, Agilent) with the mobile phases A (0.1 % TFA in water) and B (Acetonitrile) at 1 mL·min⁻¹ flow rate. Analytes were detected at 340 nm and eluted with the following gradient: starting from 90 to 80% A from 0 to 17 min, then from 80 to 60% A from 17 to 20 min, the mobile phase was maintained at 60% from 20 to 30 min, restored to initial conditions 90% A in 1 min and kept the mobile phase at 90% from 31 to 40 min. The conversion degree of L-Alanine was calculated by fitting the peak's area with a calibration curve.

## Competitive side reaction catalytic assay

A competition test[36] was designed to assess a potential substrate channeling effect[60].

Pure NADH oxidase (NOX) from *Thermus thermophilus*[61] was added as a competitor to the multi-enzymatic reaction catalyzed by the scaffolded FDH1 AlaDH3 enzymes. NOX as AlaDH3 required NADH as a cofactor, therefore the activity of NOX will be a reporter of the capability NADH accumulation as NOX will use this cofactor as substrate concomitantly and stoichiometrically producing $H_2O_2$, For the competitive catalytic assay, FDH1 at 0.18 μM, AlaDH3 at 0.18 μM, and an excess of NOX (4.4 μM) were mixed in 1 mL of reaction mixture composed of 75 mM pyruvate, 100 mM sodium formate, 500 mM ammonium chloride, 0.5 mM NADH, 0.15 mM FAD+, 0.1 mg/mL HRP, and 0.05 mM Amplex Red (AR) in pure nano water. The oxidation of the cofactor NADH to NAD+ related to the activity of both AlaDH3 and NOX was measured at 340 nm. The activity of NOX was determined by the $H_2O_2$ formation through a coupled assay using Horseradish peroxidase (HRP) and the oxidation of Amplex red as reporter by monitoring the absorbance of resorufin at 560 nm. The specific enzymatic activity

of the reactions at each wavelength were determined spectrophotometrically using Synergy H1 Hybrid Multi-Mode Microplate Reader from BioTeK Instrument in 96-well UV–Vis transparent plates.

## Isotope enrichment and dilution assay: deuterated and non-deuterated L-Alanine product formation

Deuterated and non-deuterated L-Alanine were synthesized in batch incubating scaffolded AlaDH3 and FDH1 at 1:1 molar ratio at 0.18 μM protein concentration of each in 1 mL of the reaction mixture composed by 75 mM pyruvate, 75 mM deuterated sodium formate, 25 mM sodium formate, 500 mM ammonium chloride, 0.5 mM NADH, 4.4 μM NOX, and 0.15 mM FAD$^+$ in nano pure water. Reactions were maintained under orbital agitation at 500 rpm and 25 °C for 1, 2, 4, 8, and 24 h. At different time points samples were withdrawn and analyzed by HPLC-MS to quantify the isotopic abundance of the produced L-Alanine-d-2. Reactions were stopped by tangential filtration at 13100 rcf using an Amicon Ultra-0.5 centrifugal filter units. The reaction samples at different points were diluted and then derivatized by Dansyl method[62] prior analysis by UPLC-MS (Ultra-Performance Liquid Chromatography-Mass spectrometer). UPLC was performed in a Waters ACQUITY UPLC system with a Acquity BEH C18 column (100 × 2,1 mm/1,7 μm). The gradient elution solvents were A (100 mM Ammonium formate in Water) and B (Acetonitrile) at a flow rate of 300 μL·min$^{-1}$, with the following gradient 80% A, up to 1% A for 28 min and back to 80% A for the remaining two minutes for a total duration of 30 min. The conversion degree was determined for deuterated or non-deuterated L-Alanine calculating the abundance of both hydrogen isotopes in the analyzed systems by mass spectrometry detection carried out using a time-of-flight mass spectrometer (ESI-TOF) LCT Premier XE from Waters (Milford, MA, USA) with an electrospray ionization source, working in positive/W mode. The MS range acquired was between $m/z$ 50–1000. The capillary and cone voltages were set at 3000 and 50 V, respectively. Desolvation gas temperature was 300 °C and source temperature was 120 °C. The desolvation gas flow was set at 600 L·h$^{-1}$ and cone gas flow was set at 50 L·h$^{-1}$. For quantification and data analysis, Masslynx v4.1 software was used (Waters, Milford, MA, USA). Dansyl derivatized L-Alanine detected with a mass of 323 g·mol$^{-1}$ and derivatized deuterated L-Alanine detected with a mass of 324 g·mol$^{-1}$.

## Fluorescence confocal microscopy

For imaging the distribution of the enzymes and the potential colocalization on solid beads fluorescence confocal microscopy was used. The FDH1 and AlaDH3 tagged enzymes were labeled by mixing protein solutions in PBS buffer (NaCl 150 mM, phosphate buffer 50 mM pH: 7.4) with Alexa-Fluor 647 (AF647) or Alexa-Fluor 488 (AF488) dye solutions in DMSO (1:3 molar ratio), respectively. The reactions were incubated 1 h in darkness with agitation at 25 °C. The excess of dyes was removed by gel filtration through a PD-10 column. Then, the labeled tagged enzymes were assembled onto the TRAP1-3 scaffold previously immobilized on two carriers functionalized with cobalt chelates (from Abts beads supplier) and cobalt chelates, positively charged amine groups and aldehydes at its surface, respectively. Finally, the complexes were immobilized on both carriers for 1 h in agitation at 25 °C. A suspension of the beads (1:200) was analyzed by confocal microscopy using a Zeiss LSM 510 microscope by recording the red (λ$_{ex}$: 488 nm and emission filter LP505 for AF488) and green (λ$_{ex}$: 633 nm and emission filter LP650 for AF647) channels[63]. Confocal image processing was performed using Image J (FIJI).

## L-Alanine formation in solid-phase immobilization

Immobilization of scaffolded and free enzyme systems on a solid support was carried out using tri-functional carrier[38]. First, 95 mg of tri-functional carrier was equilibrated with 950 μL of PBS buffer (150 mM NaCl, 50 mM PB pH: 8.0) for 5 min at 25 °C in a 1 mL unpacked column. Then, 950 μL of the free or scaffolded FDH1:AlaDH3 systems at 1:1 molar with of each enzyme concentration of 2.8 μM, were incubated for 4 h at 4 °C with gentle agitation (40 rpm). After incubation, five washes of the resin with PBS buffer were performed. Next, 950 μL of 1 M glycine was added to remove any free aldehyde in the resin and incubated overnight at 4 °C with gentle agitation (40 rpm). The next day the resin was washed 5 times with PBS buffer. Subsequently, the immobilization yield of both systems was determined by measuring specific enzyme activity from initial, non-immobilized and immobilized FDH1 and AlaDH3. This measurement was performed spectrophotometrically by Synergy H1 Hybrid Multi-Mode Microplate Reader from BioTeK Instrument in 96-well UV–Vis transparent plates (as previously described in Enzyme activity measurements methods). Finally, reusability of the immobilized systems was assessed. To that aim, 50 mg of the immobilized biocatalyst (with FDH1:AlaDH3 free and scaffolded at 1:1 ratio) were incubated with a 450 μL mixture composed by 500 mM ammonium chloride, 75 mM pyruvate, 100 mM sodium formate and 0.5 mM NADH in nanopure water at pH 7.5 to achieve a final enzyme concentration of each enzyme of 0.28 μM for 24 h at 25 °C at 500 rpm. The reaction was stopped by tangential filtration at 13100 rcf using an Amicon Ultra-0.5 centrifugal filter units. This was followed by 1 wash with 450 μL of 25 mM potassium phosphate pH: 8.0. The same process was repeated for 5 consecutive cycles (24 h each cycle) to analyze the reusability of the systems. The conversion degree of L-Alanine upon each cycle was determined by HPLC and calculated by fitting the peak's area with a calibration curve.

## Benzylamine formation provided by scaffolded three-enzyme multi-enzymatic system

The production of benzylamine by the scaffolded three-enzyme system was analyzed by HPLC. Benzylamine was synthesized in batch incubating 10 μM the free and scaffolded biocatalysts (FDH1, ωTA2, and AlaDH3 at 1:1:1 molar ratio) in 400 μL of reaction mixture composed by 75 mM pyruvate, 500 mM ammonium formate, 0.1 mM PLP, 10 mM benzaldehyde and 0.5 mM NADH in nano pure water. Reactions were incubated under orbital agitation at 500 rpm and 25 °C for 2, 8, 24, and 48 h. Reaction were stopped, and reaction crude were collected at each time point by centrifuging 30 min at 13100 rcf in Amicon Ultra-0.5 Centrifugal Filter Units. Samples were analyzed using Agilent Technologies 1120 Compact LC HPLC with a 5 μm Ultrabase C18 column (4.6 × 250 mm, PurpleSeries) with mobile phases A (0.1% TFA in water) and B (acetonitrile) at a flow rate of 0.9 mL·min$^{-1}$. The analytes were detected at 245 nm and eluted with the following gradient: 80–65% A from 0 to 10 min, then 65% A from 10 to 20 min, and finally then restoring the initial conditions in 10 min followed by a 10 min equilibration at these same initial conditions. The degree of conversion of benzylamine was calculated from the peak area by using a calibration curve.

## Statistics and reproducibility

No statistical method was used to predetermine sample size. No data were excluded from the analyses. The experiments were not randomized. The Investigators were not blinded to allocation during experiments and outcome assessment.

## Reporting summary

Further information on research design is available in the Nature Portfolio Reporting Summary linked to this article.

## Data availability

The data that support the findings of this study are available within this paper and supplementary information. Source data are provided with this paper. The protein structures used in this work can be found in the

Protein Data Bank with the identifiers "3FWV [https://doi.org/10.2210/pdb3FWV/pdb]"; "5DNA [https://doi.org/10.2210/pdb5DNA/pdb]"; and "1PJB [https://doi.org/10.2210/pdb1PJB/pdb]". Source data are provided with this paper.

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

## Acknowledgements

We would like to thank the financial support provided by Era-CoBioTech (Project ID: 61 HOMBIOCAT/ PCI2018-092984, F.L.-G. and A.L.C.), by European Research Council (818089, F.L-G and 648071, A.L.C.) and by the Agencia Estatal de Investigación, Spain (PID2019-111649RB-I00, A.L.C.). This work was performed under the Maria de Maeztu Units of Excellence Program from the Spanish State Research Agency Grant MDM-2017-0720 (CIC biomaGUNE). We thank Dr. J. Calvo for support with mass spectrometry and Dr. Irantzu Llanera for support circular dichroism and confocal measurements at CIC biomaGUNE. We thank the Regan Laboratory for providing the plasmids encoding the TRAP domains and Prof. Lynne Regan for early discussions about the idea.

## Author contributions

A.L.-F. performed most of the experimental work and data analysis. S.V.-L. designed and performed together with A.L.-F. the competitive assays and performed and supervised the HPLC analysis. J.S.-A. contributed to the confocal imaging of the solid beads and the data analysis related to this experiment. A.L.-F., S.V.-L., F.L.-G., and A.L.C participated in the writing and revision of the original manuscript. F.L.-G. and A.L.C conceived the idea and supervised the project. F.L.-G. and A.L.C acquired funding to develop the project. All authors participated in the revision of the manuscript and approved the final version.

## Competing interests

The authors declare no competing interests.
