## [Peer Review File · Nature Communications]

Reviewers' comments:

Reviewer #1 (Remarks to the Author):

The manuscript "Engineered repeat proteins as scaffolds to assemble multi-enzyme systems for efficient cell-free biosynthesis" by Ldesma-Fernandez et al is an interesting study in the field of spatially organizing enzymes using a protein scaffold to improve the kinetics of a chemical reaction (as compared to the same enzymes performing the reaction without a scaffold). The authors develop a scaffold that has not been used for this purpose so far, show that it complies with important criteria such as practicality and orthogonality of different binding signals, and then study and try to explain the effect of immobilizing enzymes on this scaffold. They try to demonstrate that the improved kinetics has to do with some form of "channeling" of NADH from one enzyme to its neighbor on the scaffold, and show interesting isotope enrichment data to experimentally prove their point. Finally, they expand the system from two to three enzymes and produce benzylamine with this system starting from alanine, formate, benzaldehyde, and NAD.

The topic is timely and the chosen scaffold attractive. The authors also are quite careful in characterizing the system in quite some detail. However, I doubt that this manuscript is of sufficient interest for the diverse readership of Nat Comms, for the following reasons:

1) The authors make their main argument that their systems allows them the control to actually have a look at the mechanism behind "channeling". I cannot see how the specific system they apply is really advancing the (supposedly) unresolved issues behind the mechanism of channeling. The authors make a nice experiment using deuterated NADH to illustrate that there is a direct path from product of reaction 1 to substrate of reaction 2, but it remains unclear to me why this result is a specific property of the system, as opposed to the various other systems that have been applied before that afford spatial control.

2) The authors discuss the unexplained aspects of channeling in the introduction, which is fine, but it remains unclear what the contribution of their experiment to this debate is. Which of the possible explanations discussed in the introduction is refuted (which has not been refuted before, eg by Hess)?

3) The authors undertake some careful characterizations, but in the end I am more confused than enlightened: They go through some lengths to confirm that homomultimeric enzymes assemble as such on their scaffold. Then they show in Fig. 1 the monomeric version – why? Later, they report loadings of "FDH1 and AlaDH3 per mass of carrier was 4.52 mg x g⁻¹ and 4.45 mg x g⁻¹" – but FDH has a dimeric mass of 95 kDa and AlaDH a hexameric mass of 266 kDa), yet still the authors claim a molar ratio of 1:0.34:0.34, suggesting that here they talk about monomers. At another point they claim that "The FDH1/AlaDH3@TRAP assembly showed a major chromatographic peak corresponding a molecular mass of approx. 370 kDa, which demonstrates that the complex stoichiometry is 1:1:1 according to the sequence-based molecular weights of TRAP1-3, FDH1, and AlaDH3", suggesting that a molar ratio of 1:1:1 means dimer and trimer. This ambiguity permeates the entire text and makes following the results difficult.

4) After going through the more fundamental/mechanistic aspects of the system, the authors expand the system to 3 enzymes and successfully generate benzylamine – at a level of 10 mM. All of this is interesting, but it is not clear how useful the system at this scale is.

In summary, I agree that the authors report a very interesting system, but I do not think that either the conceptual or the practical advance is large enough to justify publication in Nat Comms.

Other points:

Figure 1c: I don't understand the legend of the horizontal axis – that's pretty much the same as for the vertical axis – on purpose?

Reviewer #2 (Remarks to the Author):

The authors report on the assembly of multiple enzymes as a self-sufficient scaffold via affinity protein. Through the close proximity of the catalysts, the authors show an increased efficiency of the biotransformation and cofactor and by-product recycling.

The overall concept is very interesting but in my opinion the paper requires extensive rewriting.

The introduction is extremely lengthy, much of it could be shortened to bring the information directly relevant to the paper much sooner (for me this starts around line 128, page 3). Likewise, the results are almost integrating an experimental (overly technical) aspect which makes the manuscript challenging to follow (as in page 12 line 45: reaction set up details are already in the figures and experimental, I do not see why it needs to be repeated, or throughout the paper the specifics of the SEC analysis can be omitted in the main text and left in the supporting file) and I think for example the whole experiment with the deuterated formate could be removed from the main manuscript and relegated in supporting info, limiting to a brief reference in the main text that the direct channeling of NADH has been confirmed through isotopic enrichment experiments.

I would also argue that this is by no means a simple methodology as the authors state in their conclusion, the amount of protein engineering and the complexity of the validation of the assembled system is important. I would not think this is directly applicable to these systems either, it would have to be specifically designed for a new set of enzymes also.

I would like to be clear that I do think this work is excellent, but it is poorly presented and would greatly benefit for a full revision to make it relevant to the general public of Nat Comm and not just to the specialists.

Response to Reviewers:

Reviewers' comments:

Reviewer #1 (Remarks to the Author):

The manuscript "Engineered repeat proteins as scaffolds to assemble multi-enzyme systems for efficient cell-free biosynthesis" by Ldesma-Fernandez et al is an interesting study in the field of spatially organizing enzymes using a protein scaffold to improve the kinetics of a chemical reaction (as compared to the same enzymes performing the reaction without a scaffold). The authors develop a scaffold that has not been used for this purpose so far, show that it complies with important criteria such as practicality and orthogonality of different binding signals, and then study and try to explain the effect of immobilizing enzymes on this scaffold. They try to demonstrate that the improved kinetics has to do with some form of "channeling" of NADH from one enzyme to its neighbor on the scaffold, and show interesting isotope enrichment data to experimentally prove their point. Finally, they expand the system from two to three enzymes and produce benzylamine with this system starting from alanine, formate, benzaldehyde, and NAD.

The topic is timely and the chosen scaffold attractive. The authors also are quite careful in characterizing the system in quite some detail.

Response: We thank the reviewer for the positive comments, and we are glad that the reviewer appreciates the relevance of our work, including the attractiveness of the selected scaffold, and the detailed characterization of the system.

However, I doubt that this manuscript is of sufficient interest for the diverse readership of Nat Comms, for the following reasons:

1) The authors make their main argument that their systems allows them the control to actually have a look at the mechanism behind "channeling". I cannot see how the specific system they apply is really advancing the (supposedly) unresolved issues behind the mechanism of channeling. The authors make a nice experiment using deuterated NADH to illustrate that there is a direct path from product of reaction 1 to substrate of reaction 2, but it remains unclear to me why this result is a specific property of the system, as opposed to the various other systems that have been applied before that afford spatial control.

Response: We apologize for any confusion that may have arisen from the previous version of the manuscript regarding the novelty and advancement in the field of substrate channelling. We have made significant revisions to the manuscript, focusing on highlighting the primary achievement of our work, which is the successful combination of spatial organization and specific capture of reaction intermediates through scaffold design (see the revised manuscript, specifically page 3 introduction, pages 4-7 results, and page 19 conclusion section). This approach demonstrates that the mechanism of channelling primarily involves the increase in the local concentration of the intermediates.

We are happy that the reviewer appreciates the experiment using deuterated formate, which clearly shows the direct NADH path from reaction 1 to reaction 2. This experiment, together with the competition assays, provides unequivocal evidence of substrate channelling in our system.

We are confident that the revised version of the manuscript effectively conveys the key contribution of our work to the field.

2) The authors discuss the unexplained aspects of channeling in the introduction, which is fine, but it remains unclear what the contribution of their experiment to this debate is. Which

of the possible explanations discussed in the introduction is refuted (which has not been refuted before, eg by Hess)?

Response: As indicated above, we have focused both the introduction and the description of the results on the main contributions of our work. We show, for first time, how an engineered protein scaffold can promote efficient substrate channeling through enzyme spatial arrangement and selective interaction with the reaction intermediates. The main role of the increased local concentration of intermediates in the mechanism of channelling is unravelled.

3) The authors undertake some careful characterizations, but in the end I am more confused than enlightened: They go through some lengths to confirm that homomultimeric enzymes assemble as such on their scaffold. Then they show in Fig. 1 the monomeric version – why? Later, they report loadings of “FDH1 and AlaDH3 per mass of carrier was 4.52 mg x g⁻¹ and 4.45 mg x g⁻¹” – but FDH has a dimeric mass of 95 kDa and AlaDH a hexameric mass of 266 kDa), yet still the authors claim a molar ratio of 1:0.34:0.34, suggesting that here they talk about monomers.

At another point they claim that “The FDH1/AlaDH3@TRAP assembly showed a major chromatographic peak corresponding a molecular mass of approx. 370 kDa, which demonstrates that the complex stoichiometry is 1:1:1 according to the sequence-based molecular weights of TRAP1-3, FDH1, and AlaDH3”, suggesting that a molar ratio of 1:1:1 means dimer and trimer. This ambiguity permeates the entire text and makes following the results difficult.

Response: We are sorry that this part was not clear enough in the previous version. To address this issue, we have now simplified the description of the assemblies and provided additional clarification by giving the stoichiometry of the multimers, instead just the monomers, of the enzymes. Additionally, we have included a schematic representation of the pursued assembly (Figure 1B) that reflects the stoichiometry achieved by controlling the protein concentration and the assembly sequence. We have also provided experimental evidence of the assembly size and composition by size exclusion chromatography analysis. We believe that the description of the assembly is now consistent throughout the text.

4) After going through the more fundamental/mechanistic aspects of the system, the authors expand the system to 3 enzymes and successfully generate benzylamine – at a level of 10 mM. All of this is interesting, but it is not clear how useful the system at this scale is.

Response: We believe that expanding the system to include three enzyme is important to demonstrate the versatility and applicability of the system for templating more complex multi-enzyme systems. The specific reaction that was tested shows the potential of the system to not only recycle redox co-factors but also amine donors. The scale at which this was demonstrated allowed us reaching a 60% benzylamine yield using ammonia as ultimate donor. The relevance of this cascade is that the L-Alanine is produced in situ with a low transient concentration to which the free transaminase would not work. However, as the transaminase is in the physical proximity of the L-Alanine dehydrogenase where the L-Alanine is produced, we suggest that this local production of the amine donor for the transaminase makes this cascade possible. To note, high excess (50 equivalents) of L-Ala are required to make the transaminases to work, shifting the thermodynamic equilibrium toward the amination reaction. Transaminases rarely work starting from pyruvate and ammonium. So the assembly of these three enzymes cascade allows bringing the transaminases out of the equilibrium. Of course, we acknowledge that the substrate load must be increased to make this process attractive for the industry, but we believe that our work proves the concept that the transaminase equilibrium can be shifted by spatial arrangement of enzymes.

In summary, I agree that the authors report a very interesting system, but I do not think that either the conceptual or the practical advance is large enough to justify publication in Nat Comms.

Response: We thank the reviewer for taking the time to review our work and for considering it interesting. We believe that the revised version of our manuscript addresses the concerns and limitations of the previous version, and more convincingly conveys the novelty and progress of our work. We hope that upon review of the revised manuscript, the reviewer will find that the conceptual and practical advances justify publication in Nature Communications.

Other points:

Figure 1c: I don't understand the legend of the horizontal axis – that's pretty much the same as for the vertical axis – on purpose?

Response: The legend was intended to emphasize the results. We have edited the figure and removed the additional legend for clarity.

Reviewer #2 (Remarks to the Author):

The authors report on the assembly of multiple enzymes as a self-sufficient scaffold via affinity protein. Through the close proximity of the catalysts, the authors show an increased efficiency of the biotransformation and cofactor and by-product recycling.

The overall concept is very interesting but in my opinion the paper requires extensive rewriting.

The introduction is extremely lengthy, much of it could be shortened to bring the information directly relevant to the paper much sooner (for me this starts around line 128, page 3).

Response: We would like to thank the reviewer for the time to review our work. We appreciate that the reviewer finds the main concept of the manuscript interesting. Following the reviewer's suggestions, we have revised the introduction to be more concise and focused, bringing the relevant information to the forefront, and ensuring that the paper goes straight to the point. We hope that now the reviewer finds our revised version more focused and engaging.

Likewise, the results are almost integrating an experimental (overly technical) aspect which makes the manuscript challenging to follow (as in page 12 line 45: reaction set up details are already in the figures and experimental, I do not see why it needs to be repeated, or throughout the paper the specifics of the SEC analysis can be omitted in the main text and left in the supporting file) and I think for example the whole experiment with the deuterated formate could be removed from the main manuscript and relegated in supporting info, limiting to a brief reference in the main text that the direct channeling of NADH has been confirmed through isotopic enrichment experiments.

Response: We appreciate the reviewer's suggestion to remove technical aspects from the results section and have revised it accordingly, making this section more concise (see the revised manuscript). However, after careful consideration of the comment regarding the deuterated formate experiment, we believe that this example is crucial to demonstrating the direct channeling of NADH, as suggested by Reviewer 1, and provides essential supporting evidence for the claims made in the main text. Thus, after careful consideration of your comments, we have decided to keep this description within the main text. To simplify this section, we have also reduced the technical aspects. As Nature Communication is a journal for a wide audience in sciences, we humbly believe that keeping this technical information

about the isotopic enrichment will help to understand the results to readers who are not familiarized with this methodology.

I would also argue that this is by no means a simple methodology as the authors state in their conclusion, the amount of protein engineering and the complexity of the validation of the assembled system is important. I would not think this is directly applicable to these systems either, it would have to be specifically designed for a new set of enzymes also.

Response: We thank the reviewer for this comment and would like to expand the explanation about the current methodology. We believe that the methodology is relatively simple and modular compared to other current approaches based on the following facts: 1) The scaffolds are exquisitely orthogonal and; 2) the peptide tagging approach is simple since only short 10-20 amino acid sequences are needed to anchor enzymes to the scaffoldin. This approach is genetically programmed very easily as we constructed a set of plasmids for block cloning and tagging of any enzyme; 3) the peptide tagging has not resulted in any major defect on the protein activity for all the systems tested up to date; 4) the cofactor interactions are driven by easily encodable electrostatic interactions; and 5) the methodology also involves a straightforward incubation process for assembling the components, which makes it relatively simple and scalable for use in large-scale applications.

Additionally, the authors provide experimental data and validation of the methodology using multiple enzymes, indicating that it has broad applicability and potential for use in a range of different enzyme systems. Here, three enzymes with a high structural diversity both at tertiary and quaternary levels can be assembled productively through the same approach. Thus, we consider this approach as quite generalizable for other multi-enzyme systems.

I would like to be clear that I do think this work is excellent, but it is poorly presented and would greatly benefit for a full revision to make it relevant to the general public of Nat Comm and not just to the specialists.

Response: We thank the reviewer for taking the time to review our work and for considering it excellent. We believe that the revised version of our manuscript addresses the concerns and limitations of the previous version, and more convincingly conveys the novelty and progress of our work. We hope that upon review of the revised manuscript, the reviewer will find relevant to the general public and justify publication in Nature Communications.

REVIEWERS' COMMENTS

Reviewer #1 (Remarks to the Author):

The revised version of «Engineered repeat proteins as scaffolds to assemble multi-enzyme systems for efficient cell-free biosynthesis» by Ledesma-Fernandez et al has gained much in clarity of the presentation, and I am still very happy to confirm that I think that the experiments were performed with much diligence and creativity.

Regarding my earlier points: The authors have addressed my concerns about which part of the “scaffolded vs non-scaffolded” discussion they are addressing and how. This part has become much clearer and much more convincing. However, as the authors discuss themselves, the effect of retaining a charged intermediate close to the scaffold because of the electrostatic interactions is not new – it has been shown with a fusion enzyme, for example. Here, it is shown for a scaffold, and I do not fully understand why this is a major advance. However, I do not see this point any longer as critical to the final assessment of the paper.

The authors also clarified my concerns about potentially confusing use of the terms “monomeric” etc. Also this part is now much clearer.

I still struggle with the wider impact of the demonstrate examples. For the two enzyme system, the authors point towards Figure S4, in which they show that the advantage of the scaffolded enzyme set over the free enzyme set (in terms of reaction rates) is a function of the NADH concentration and reaches a maximum for 0.5 mM. However, the benefit is gone at 1 mM NADH, so a simple and minor change in the concentration of a critical component removes the effect entirely. This might be different in the 3-enzyme case (as the formation of alanine from pyruvate and an excess of ammonia might be limited by the position of the equilibrium, and therefore there could be an intrinsic reason why scaffolding is required as the performance improvement cannot be achieved in another way), but this argument is not used in the manuscript (as far as I can tell). As it is, the three-enzyme system is a nice example, but the overall performance in itself is a bit underwhelming.

However, overall I think that the present manuscript describes a remarkable piece of work, that might find its impact in the new system (TRAP), in the insights that are provided regarding the mechanism of the channeling, and maybe also for the set-up of complex systems in which the accumulation of intermediates is limited by the equilibrium position (maybe, after a revision of the corresponding introductory text at the beginning of the section “Amine biosynthesis catalyzed by a TRAP-scaffolded tri-enzyme system”). Therefore, I recommend acceptance after minor revision.

Reviewer #2 (Remarks to the Author):

The authors have significantly improved the structure of the manuscript that now reads much smoother and to the point. I appreciate also the additional aspects clarified in this version (i.e. the reason why the system could be more broadly applicable), and the streamlining of the results to avoid too much technical details in the main text.

The work is solid and very carefully validated by extensive experiments which indeed support the claims of the authors (the benefits of precise scaffolding technology).

I feel the authors have also addressed the criticism raised by the other reviewer so for me this manuscript is certainly worth considering for publication.

There are a number of very minor linguistic inaccuracies which I'm sure will be corrected in the editing process.

REVIEWERS' COMMENTS

Reviewer #1 (Remarks to the Author):

The revised version of «Engineered repeat proteins as scaffolds to assemble multi-enzyme systems for efficient cell-free biosynthesis» by Ledesma-Fernandez et al has gained much in clarity of the presentation, and I am still very happy to confirm that I think that the experiments were performed with much diligence and creativity.

Response: We thank the reviewer for the positive comments. We are pleased to hear that the reviewer appreciates the relevance and robustness of our work.

Regarding my earlier points: The authors have addressed my concerns about which part of the “scaffolded vs non-scaffolded” discussion they are addressing and how. This part has become much clearer and much more convincing.

Response: We thank the reviewer for the revision. We are glad that our thorough revisions following the reviewer’s suggestion have significantly clarified the manuscript.

However, as the authors discuss themselves, the effect of retaining a charged intermediate close to the scaffold because of the electrostatic interactions is not new – it has been shown with a fusion enzyme, for example. Here, it is shown for a scaffold, and I do not fully understand why this is a major advance. However, I do not see this point any longer as critical to the final assessment of the paper.

Response: We thank the reviewer for the feedback on this point. We agree that the effect of retaining a charged intermediate by electrostatic interactions have been shown before. However, we believe that the system developed in this work is more interesting than fusion proteins due to its modularity. This modularity allows for tuning the strength of the electrostatic interactions, or creating additional modules that do not connect enzymes, but instead sequester intermediates. We believe that this feature provides a lot of potential for the creation of a wide range of catalytic systems.

The authors also clarified my concerns about potentially confusing use of the terms “monomeric” etc. Also this part is now much clearer.

Response: We thank the reviewer for the revision. We are pleased to hear that this revised section is also clearer.

I still struggle with the wider impact of the demonstrate examples. For the two enzyme system, the authors point towards Figure S4, in which they show that the advantage of the scaffolded enzyme set over the free enzyme set (in terms of reaction rates) is a function of the NADH concentration and reaches a maximum for 0.5 mM. However, the benefit is gone at 1 mM NADH, so a simple and minor change in the concentration of a critical component removes the effect entirely.

Response: We acknowledge that we have only applied the cofactor channeling concept to NADH-dependent enzyme cascades. Nonetheless, we envision TRAP modules as scaffolds to spatially organize and channel multi-enzyme systems where the regeneration of others phosphorylated cofactors (i.e FAD, ATP...) is critical. TRAPs play a dual role in organizing the enzymes at the nanoscale and sequestering the phosphorylated cofactors, thus increasing

their local concentrations at the surrounding of the scaffolded enzymes. With these arguments, we hope to convince reviewer 1 about the broader impact of this biomolecular system for other multi-enzyme systems.

Regarding the diffusion-limited channeling being highly affected by the NADH concentration in the bulk, the reviewer 1 is totally right. Above certain concentration of NADH, the channeling effect is lost due to the molecular nature of the channeling phenomena we have artificially created. According to this, the system is saturated when the NADH bulk concentration is much larger than the K_D of TRAP towards this cofactor ($K_D = 0.038$ mM), and the maximum rate of the scaffolded system remains constant. At NADH concentration higher than 1 mM, both scaffolded and free systems are saturated, thus the diffused-limited channeling effect is vanished. On the contrary, when the NADH concentration in the bulk is very low (< 0.05 mM), the NADH concentration, even at the surrounding of the scaffolded enzymes, is still significantly lower than the K_M of the scaffolded enzymes toward their cofactor ($K_M(\text{FDH1}) = 0.34$, $K_M(\text{AlaDH3}) = 0.25$ mM), limiting the overall throughput of the cascade. In this scenario, the intrinsic binding affinity of the two enzymes dominates over channeling effect. Therefore, the channeling only dominates over the intrinsic enzyme kinetics when the cofactor concentration is significantly higher than the K_D of the TRAP towards the cofactor, but not sufficiently higher than the K_M of the enzymes towards that cofactor.

To illustrate this trade-off between the cofactor binding to the scaffold and the cofactor binding to the enzymes more clearly, we have included a saturation plot (Figure S4B) for the L-Ala synthetic rate at different NADH bulk concentrations. This plot demonstrates that the scaffolded system operates at its maximum speed at much lower concentration of NADH than the free system, supporting the benefits of the diffusion-limited channeling driven by the specific TRAP-cofactor reversible interaction. Figure S4B also reveals that the maximum rate in the cascade is achieved at 0.5 mM NADH, a concentration similar to which V_{max} of the rate-limiting enzyme FDH1 alone (Figure S3A) is reached. This finding suggests a local concentration of 0.5 mM NAD^+ in the surroundings of FDH when it is scaffolded together with its NAD^+ producing partner (AlaDH). Conversely, when these two enzymes are not scaffolded, the estimated local NAD^+ concentration that FDH encounters is approximately half the NADH concentration we supply to the bulk, as shown in Figure S4B and Figure S3A.

This might be different in the 3-enzyme case (as the formation of alanine from pyruvate and an excess of ammonia might be limited by the position of the equilibrium, and therefore there could be an intrinsic reason why scaffolding is required as the performance improvement cannot be achieved in another way), but this argument is not used in the manuscript (as far as I can tell). As it is, the three-enzyme system is a nice example, but the overall performance in itself is a bit underwhelming.

Response: We really appreciate the argument given by reviewer 1 as it is very useful for the discussion of the 3-enzyme system. We have included this argument in the discussion section of the manuscript related to the effects of the scaffolding of the 3-enzyme system (Page 18). However, we do not fully agree with the reviewer 1 about the fact that the overall performance of the scaffolded 3-enzyme cascade is underwhelming. According to **Figure 8C**, the scaffolded system produces 4.25-fold higher benzylamine yield with a 2.36 times higher productivity than the free system. As reviewer 1 points out, this enhanced performance has been achieved starting without the amine donor for the transaminase, which dramatically hampers the transamination reactions due to thermodynamic reasons. Woodley and co-workers have deeply investigated the effect of amine donor on the thermodynamics of the transaminase reactions (<http://dx.doi.org/10.1016/j.tetasy.2015.04.006>), demonstrating that L-Ala is one of the worst amine donors in terms of thermodynamics for this reaction, although this is the natural amine donor for most transaminases. The reason behind these impaired thermodynamics is that pyruvate is an excellent amine acceptor. Therefore, when the aminated product accumulates in presence of an excess of pyruvate, it can be deaminated back by the transaminase, forming L-Alanine. Hence, achieving a 60% amination yield starting

from 0 mM L-Ala and 7.5 equivalents of pyruvate regarding benzaldehyde is quite remarkable. This study is one of the few that exploits the *in-situ* formation of the amine donor to perform a transamination reaction in presence of a potential amine acceptor that favors the equilibrium towards deamination reaction. An excess of pyruvate is typically used for deamination reaction in kinetic resolution of amines or to produce carbonyl compounds (DOI: [10.1021/acscatal.7b02686](https://doi.org/10.1021/acscatal.7b02686)). In summary, thanks to the scaffolding of a model multi-enzyme system, we have achieved significant amine yields through a thermodynamically unfavorable pathway, with similar productivity as the reported for the same reaction catalyzed by the same transaminase, yet using 10 equivalents of amine donor (L-Alanine) (doi.org/10.1016/j.ijbiomac.2020.09.003).

However, overall I think that the present manuscript describes a remarkable piece of work, that might find its impact in the new system (TRAP), in the insights that are provided regarding the mechanism of the channeling, and maybe also for the set-up of complex systems in which the accumulation of intermediates is limited by the equilibrium position (maybe, after a revision of the corresponding introductory text at the beginning of the section “Amine biosynthesis catalyzed by a TRAP-scaffolded tri-enzyme system”).

Response: We thank the reviewer for these words. Accordingly, we have added an introductory sentence in this section to highlight how channeling is performing this amination cascade starting from the ammonia/pyruvate pair as amine source (see Pag 17).

Therefore, I recommend acceptance after minor revision.

Response: Thanks for the positive feedback and the recommendation for publication.

Reviewer #2 (Remarks to the Author):

The authors have significantly improved the structure of the manuscript that now reads much smoother and to the point. I appreciate also the additional aspects clarified in this version (i.e. the reason why the system could be more broadly applicable), and the streamlining of the results to avoid to much technical details in the main text.

The work is solid and very carefully validated by extensive experiments which indeed support the claims of the authors (the benefits of precise scaffolding technology). I feel the authors have also addressed the criticism raised by the other reviewer so for me this manuscript is certainly worth considering for publication.

There a number of very minor linguistic inaccuracies which I'm sure will be corrected in the editing process.

Response: We would like to thank the reviewer for the detailed evaluation of our manuscript and previous comments that helped improve our work. We greatly appreciate the positive feedback in this revised version of the manuscript and the recommendation for publication.